# Robust Clustering using Gaussian Mixtures in the Presence of Cellwise Outliers

**Pushpendra Rajpurohit**                                                      *crz238143@iitd.ac.in*
*Centre for Applied Research in Electronics*
*Indian Institute of Technology, Delhi 110016, India*

**Petre Stoica**                                                                      *ps@it.uu.se*
*Division of Systems and Control, Department of Information Technology*
*Uppsala University, Uppsala, Sweden 75237*

**Prabhu Babu**                                                        *Prabhu.Babu@care.iitd.ac.in*
*Centre for Applied Research in Electronics*
*Indian Institute of Technology, Delhi 110016, India*

**Reviewed on OpenReview:** *https://openreview.net/forum?id=oVHPEgjdWk*

## Abstract

In this paper we propose a novel algorithm for robust estimation of Gaussian Mixture Model (GMM) parameters and clustering that explicitly accounts for cell outliers. To achieve this, the proposed algorithm minimizes a penalized negative log-likelihood function where the penalty term is derived via the false discovery rate principle. The penalized negative log-likelihood function is cyclically minimized over outlier positions and the GMM parameters. Furthermore, the minimization over the GMM parameters is done using the majorization minimization framework: specifically we minimize a tight upper bound on the negative log-likelihood function which decouples into simpler optimization subproblems that can be solved efficiently. We present several numerical simulation studies comprising experiments aimed at evaluating the performance of the proposed method on synthetic as well as real world data and at systematically comparing it with state-of-the-art robust techniques in different scenarios. The simulation studies demonstrate that our approach effectively addresses the challenges inherent in parameter estimation of GMM and clustering in contaminated data environments.

## 1 Introduction and literature

In machine learning, pattern classification, and other statistical fields, a fundamental challenge lies in estimating the parameters of the underlying distribution from observed data samples corrupted by outliers (Thomaz et al., 2004; Minh & Murino, 2017; Bois et al., 2024). For example, health monitoring systems may suffer from faulty measurements (Sangra & Codina, 2015), while socio-economic surveys might include deliberately falsified responses (Menold & Kemper, 2014). To address these issues, robust statistical methodologies have been developed to accurately estimate the parameters of the outlier corrupted data by first detecting outliers, then either down-weighting or completely eliminating them and finally estimating the parameters from the remaining uncorrupted data. The outliers may be casewise outliers (Huber, 1964; Stoica et al., 2024), or cellwise outliers (Alqallaf et al., 2009), the latter being more challenging to deal with.

The inherent complexity of real-world data often cannot be adequately captured by a single distribution. To take this complexity into account, a mixture model that is a linear combination of several basic distributions can be employed. When these component distributions are Gaussian, the resulting model is termed a Gaussian mixture model (GMM). The superposition of multiple component distributions facilitates the representation of intricate density functions, and GMMs are widely utilized to reveal possible underlying

clusters of the data samples (Bishop & Nasrabadi, 2006; Reynolds et al., 2009; Yang et al., 2012; Sharma et al., 2024; Cuesta-Albertos & Dutta, 2023).

In many applications of GMMs the data are corrupted by outliers that do not belong to any of the mixture distributions. To handle outlier corrupted data, one approach is to partially relax the normality assumption by considering heavy-tailed distributions for the components. This framework includes models such as the mixture of Student's $t$ distributions (Peel & McLachlan, 2000) and the mixtures of contaminated normal distributions (Punzo & McNicholas, 2016). To address arbitrarily distributed outliers within the model-based clustering, García-Escudero et al. (2008) introduced robust trimmed clustering (TCLUST), an approach that extends the minimum covariance determinant (MCD) method for estimating the distribution parameters using classification trimmed likelihoods. In particular, TCLUST incorporates a concentration step within its expectation maximization (EM) algorithm, analogous to the one used in the FAST-MCD procedure (Rousseeuw & van Driessen, 1999), where a fixed proportion of the observations that deviate most from the mean are considered to be outliers and are omitted from the parameter estimation step. A similar trimming method has also been implemented within a mixture model framework in Neykov et al. (2007) and García-Escudero et al. (2014). blueRecent probabilistic formulations have also incorporated adaptive weighting to enhance robustness. Wang et al. (2017) introduced a Bayesian data reweighting framework that learns per-sample weights jointly with model parameters, thereby reducing the influence of outlier observations and improving resistance to model misspecification. Similarly, Ghahramani & Jordan (1993) proposed an EM-based approach for supervised learning with incomplete data, where missing or uncertain entries are treated as latent variables, providing a probabilistic foundation relevant to handling cell-wise contamination in mixture models.

The goal of the present study is to develop an outlier-robust GMM parameter estimation approach that can detect the cell outliers. To achieve this, we employ the multiple hypothesis testing procedure of false discovery rate (FDR) to detect the outliers and employ the majorization-minimization (MM) approach to estimate the GMM parameters from the uncorrupted data. Unlike the classical EM algorithm, which necessitates the introduction of latent variables and the computation of conditional expectations, our MM-based approach for estimating the GMM parameters circumvents these steps and it is straightforward to understand and implement. The proposed method alternatingly estimates the cell outlier positions and the GMM parameters by minimizing a penalized negative log likelihood objective until a convergence criterion is met, where the penalty term is derived from the FDR principle. The EM and MM yield identical parameter updates (means, covariances, and mixture proportions) for the non-outlier case (Sahu & Babu, 2020). In the presence of outliers, we prefer MM, as given the latest estimate of outlier positions, EM is typically run until convergence but on the other hand, a single iteration of MM is sufficient to decrease the penalized negative log likelihood objective.

The remaining sections of the paper are organized as follows. In Section II, we introduce the data model and formulate the minimization problem of penalized negative log-likelihood objective function with respect to the cell outlier positions and the GMM parameters. In Section III, we present a brief overview of the MM framework and the FDR-based multiple hypothesis testing. Section IV presents the derivations of cell outlier detection in the GMM and the GMM parameter estimation via the MM approach. This section offers an in-depth, step-by-step exposition of the mathematical formulations, emphasizing the iterative optimization process that underpins our method. In Section V, we conduct an extensive simulation study on synthetic as well as real world data to evaluate the performance of the proposed framework under a variety of simulation conditions, and we benchmark its performance against state-of-the-art robust parameter estimation and clustering techniques. Finally, Section VI concludes the paper.

## 2 Problem Formulation

In this section, we discuss the GMM as well as the GMM in the presence of cell outliers, and formulate the minimization problem with respect to the cell outlier positions and the GMM parameters.

Let $\mathbf{y}(t)$ be a $p$-dimensional vector that is randomly drawn from a mixture of $K$ Gaussian distributions. The corresponding mixture distribution is given by:

$$p\left(\mathbf{y}(t); \boldsymbol{\theta}\right) = \sum_{k=1}^{K} \pi_k \, \mathcal{N}\left(\mathbf{y}(t); \boldsymbol{\mu}_k, \boldsymbol{\Sigma}_k\right), \tag{1}$$

where the parameter vector $\boldsymbol{\theta} = \{\pi_k, \boldsymbol{\mu}_k, \boldsymbol{\Sigma}_k\}_{k=1}^{K}$ comprises the mixing proportions, means, and covariance matrices of the $K$ components and $\mathcal{N}\left(\mathbf{y}(t); \boldsymbol{\mu}_k, \boldsymbol{\Sigma}_k\right)$ denotes the distribution of the $k^{\text{th}}$ component, viz:

$$\mathcal{N}\left(\mathbf{y}(t); \boldsymbol{\mu}_k, \boldsymbol{\Sigma}_k\right) \triangleq \frac{1}{\sqrt{(2\pi)^p |\boldsymbol{\Sigma}_k|}} \exp\left(-\frac{1}{2}\left(\mathbf{y}(t) - \boldsymbol{\mu}_k\right)^{\top} \boldsymbol{\Sigma}_k^{-1} \left(\mathbf{y}(t) - \boldsymbol{\mu}_k\right)\right). \tag{2}$$

In many practical scenarios, the vector $\mathbf{y}(t)$ may be partially corrupted by cell outliers. To address this situation, we introduce a binary vector $\mathbf{b}(t)$ associated with $\mathbf{y}(t)$: the element $b_j(t)$ of the binary vector $\mathbf{b}(t)$ is set to 1 if the $j^{\text{th}}$ element of $\mathbf{y}(t)$ is uncorrupted and to 0 if the $j^{\text{th}}$ element of $\mathbf{y}(t)$ is contaminated. Based on the binary vector $\mathbf{b}(t)$, we construct a binary selection matrix $\mathbf{B}_t \in \mathbb{R}^{p \times p_t}$ where $p_t$ is the number of uncorrupted elements in $\mathbf{y}(t)$. The matrix $\mathbf{B}_t$ is defined so that each column contains exactly one entry equal to 1 corresponding to the uncorrupted elements of $\mathbf{y}(t)$, with all other entries set to 0. For example, if

$$\mathbf{y}(t) = [y_1(t), y_2(t), y_3(t)]^{\top}$$

and the corresponding indicators are $b_1(t) = 1$, $b_2(t) = 1$, and $b_3(t) = 0$, then the matrix $\mathbf{B}_t$ is given by:

$$\mathbf{B}_t = \begin{bmatrix} 1 & 0 \\ 0 & 1 \\ 0 & 0 \end{bmatrix}.$$

This structure of $\mathbf{B}_t$ ensures that $\mathbf{B}_t^{\top} \mathbf{y}(t)$ extracts the uncorrupted elements of $\mathbf{y}(t)$, and thus $\mathbf{B}_t^{\top} \boldsymbol{\Sigma}_k \mathbf{B}_t$ is the covariance matrix corresponding to these elements. Since $\{\mathbf{B}_t\}$ can be directly obtained from $\{\mathbf{b}(t)\}$ and vice versa, the two notations are equivalent. We employ both representations because $\mathbf{B}_t$ facilitates the expression of the uncorrupted data and its associated covariance (i.e., $\mathbf{B}_t^{\top} \mathbf{y}(t)$ and $\mathbf{B}_t^{\top} \boldsymbol{\Sigma}_k \mathbf{B}_t$), while $\mathbf{b}(t)$ is a more convenient notation when estimating the positions of the cell outliers.

In the presence of cell outliers, if an observed sample $\mathbf{y}(t)$ is drawn from the GMM then the distribution corresponding to the uncorrupted elements of the data is given by:

$$p\left(\mathbf{y}(t); \boldsymbol{\theta}, \mathbf{b}(t)\right) = \sum_{k=1}^{K} \pi_k \mathcal{N}\left(\mathbf{y}(t); \boldsymbol{\mu}_k, \boldsymbol{\Sigma}_k, \mathbf{b}(t)\right), \tag{3}$$

where:

$$\mathcal{N}\left(\mathbf{y}(t); \boldsymbol{\mu}_k, \boldsymbol{\Sigma}_k, \mathbf{b}(t)\right) \triangleq \frac{1}{\sqrt{(2\pi)^{p_t} \left|\mathbf{B}_t^{\top} \boldsymbol{\Sigma}_k \mathbf{B}_t\right|}} \exp\left(-\frac{1}{2}\left(\mathbf{y}(t) - \boldsymbol{\mu}_k\right)^{\top} \mathbf{B}_t \left(\mathbf{B}_t^{\top} \boldsymbol{\Sigma}_k \mathbf{B}_t\right)^{-1} \mathbf{B}_t^{\top} \left(\mathbf{y}(t) - \boldsymbol{\mu}_k\right)\right), \tag{4}$$

and $p_t = \sum_{j=1}^{p} b_j(t)$.

Assume now that we have a cell-corrupted dataset $\mathcal{D} = \{\mathbf{y}(1), \cdots, \mathbf{y}(N)\}$ consisting of $N$ independent and identically distributed samples drawn from the GMM in (3). The objective is to estimate both the model parameters $\boldsymbol{\theta}$ and the binary variables $\{\mathbf{b}(t)\}$. To facilitate this estimation, we define the following function that will be frequently referenced in subsequent derivations:

$$g_{tk}\left(\boldsymbol{\theta}_k, \mathbf{b}(t)\right) \triangleq \log\left(\pi_k \mathcal{N}\left(\mathbf{y}(t); \boldsymbol{\mu}_k, \boldsymbol{\Sigma}_k, \mathbf{b}(t)\right)\right), \tag{5}$$

where $\boldsymbol{\theta}_k \triangleq \{\pi_k, \boldsymbol{\mu}_k, \boldsymbol{\Sigma}_k\}$. Note that (5) can be written as:

$$g_{tk}\left(\boldsymbol{\theta}_k, \mathbf{b}(t)\right) = \log \pi_k - \frac{1}{2}\left(\mathbf{y}(t) - \boldsymbol{\mu}_k\right)^{\top} \mathbf{B}_t \left(\mathbf{B}_t^{\top} \boldsymbol{\Sigma}_k \mathbf{B}_t\right)^{-1} \mathbf{B}_t^{\top} \left(\mathbf{y}(t) - \boldsymbol{\mu}_k\right) - \frac{1}{2}\log\left|\mathbf{B}_t^{\top} \boldsymbol{\Sigma}_k \mathbf{B}_t\right| - \frac{p_t}{2}\log(2\pi). \tag{6}$$

Also note that:

$$\pi_k \mathcal{N}\left(\mathbf{y}(t); \boldsymbol{\mu}_k, \boldsymbol{\Sigma}_k, \mathbf{b}(t)\right) = e^{g_{tk}(\boldsymbol{\theta}_k, \mathbf{b}(t))}. \tag{7}$$

For the dataset $\mathcal{D}$, the likelihood function corresponding to the uncorrupted data can be expressed as:

$$L\left(\boldsymbol{\theta}; \mathcal{D}, \{\mathbf{b}(t)\}\right) \triangleq \prod_{t=1}^{N} p\left(\mathbf{y}(t); \boldsymbol{\theta}, \mathbf{b}(t)\right). \tag{8}$$

Therefore the negative log-likelihood is given by:

$$l\left(\boldsymbol{\theta}; \mathcal{D}, \{\mathbf{b}(t)\}\right) \triangleq -\log L\left(\boldsymbol{\theta}; \mathcal{D}, \{\mathbf{b}(t)\}\right) = -\sum_{t=1}^{N} \log\left(\sum_{k=1}^{K} e^{g_{tk}(\boldsymbol{\theta}_k, \mathbf{b}(t))}\right). \tag{9}$$

For later use, we also define the negative log-likelihood of the $t^{\text{th}}$ observation $\mathbf{y}(t)$ as follows:

$$l_t\left(\boldsymbol{\theta}, \mathbf{y}(t), \mathbf{b}(t)\right) \triangleq -\log\left(\sum_{k=1}^{K} e^{g_{tk}(\boldsymbol{\theta}_k, \mathbf{b}(t))}\right) \tag{10}$$

Minimizing (9) will flag too many cells as outliers. To prevent this from happening we add a penalty term to (9) which penalizes the number of outliers. The problem of estimating the GMM parameters $\boldsymbol{\theta}$ and the binary variables $\{\mathbf{b}(t)\}$ will thus be formulated as:

$$\begin{aligned} &\underset{\{N_i\}, \boldsymbol{\theta}_k, \{\mathbf{b}(t)\}}{\text{minimize}} && l\left(\boldsymbol{\theta}; \mathcal{D}, \{\mathbf{b}(t)\}\right) + \sum_{i=1}^{p} \sum_{t=1}^{N_i} \eta_t, \\ &\text{subject to} && \boldsymbol{\pi}^T \mathbf{1} = 1, \quad \boldsymbol{\pi} \succeq 0, \quad \boldsymbol{\Sigma}_k \succ 0 \quad \forall k, \end{aligned} \tag{11}$$

where $N_i = N - \sum_{t=1}^{N} b_i(t)$ denotes the number of corrupted cells corresponding to the $i^{\text{th}}$ variable across all observations and $\eta_t$ is a penalty factor whose choice will be based on the FDR as discussed later.

## 3 Preliminaries

In this section, we provide concise descriptions of the MM procedure (Sun et al., 2016) and the FDR principle (Stoica & Babu, 2022), which will be useful in the algorithm development.

### 3.1 Majorization Minimization

Consider the following minimization problem:

$$\underset{\boldsymbol{\theta} \in \boldsymbol{\Theta}}{\text{minimize}} \quad f(\boldsymbol{\theta}), \tag{12}$$

where $\boldsymbol{\theta}$ denotes the optimization variable and $\boldsymbol{\Theta}$ represents the feasibility set.

Let $\boldsymbol{\theta}^z \in \boldsymbol{\Theta}$ denote the current estimate of $\boldsymbol{\theta}$ at the $z^{\text{th}}$ iterative step. A surrogate function $g_f(\boldsymbol{\theta} \mid \boldsymbol{\theta}^z)$ is said to majorize the original objective function $f(\boldsymbol{\theta})$ at $\boldsymbol{\theta}^z$ if (Sun et al., 2016; Hunter & Lange, 2004):

$$f(\boldsymbol{\theta}) \leq g_f(\boldsymbol{\theta} \mid \boldsymbol{\theta}^z) \quad \forall \boldsymbol{\theta} \in \boldsymbol{\Theta}, \tag{13}$$

and

$$f(\boldsymbol{\theta}^z) = g_f(\boldsymbol{\theta}^z \mid \boldsymbol{\theta}^z). \tag{14}$$

Instead of directly minimizing $f(\boldsymbol{\theta})$, the MM procedure minimizes the surrogate function $g_f(\boldsymbol{\theta} \mid \boldsymbol{\theta}^z)$. The minimizer of this surrogate function is then taken as the updated estimate $\boldsymbol{\theta}^{z+1}$, i.e.,

$$\boldsymbol{\theta}^{z+1} = \arg \underset{\boldsymbol{\theta} \in \boldsymbol{\Theta}}{\text{minimize}}\, g_f(\boldsymbol{\theta} \mid \boldsymbol{\theta}^z). \tag{15}$$

The update in (15) guarantees a non-increasing sequence of objective function values:

$$f(\boldsymbol{\theta}^{z+1}) \overset{(13)}{\leq} g_f(\boldsymbol{\theta}^{z+1} \mid \boldsymbol{\theta}^z) \overset{(15)}{\leq} g_f(\boldsymbol{\theta}^z \mid \boldsymbol{\theta}^z) \overset{(14)}{=} f(\boldsymbol{\theta}^z). \tag{16}$$

Thus, starting from an initial point $\boldsymbol{\theta}^0 \in \boldsymbol{\Theta}$, the MM procedure generates a sequence $\{\boldsymbol{\theta}^z\}$ that monotonically decreases the objective function. For further details on the construction of surrogate functions, readers are referred to Sun et al. (2016).

## 3.2 False Discovery Rate for Multiple Hypothesis Testing

For outlier detection, we utilize the multiple hypothesis testing procedure of FDR (Benjamini & Hochberg, 1995; Stoica & Babu, 2022). Let $\{H_t\}_{t=1}^N$ denote the set of $N$ null hypotheses and $\{T(t)\}_{t=1}^N$ be the corresponding test statistics. The multiple hypothesis testing, which is usually preferred to individual testing for large $N$, (Stoica et al., 2024) is often performed by controlling the FDR (Benjamini & Hochberg, 1995). Let $\gamma_t$ denote the individual false alarm probability (also called the significance level):

$$\gamma_t = \mathrm{prob}(\mathrm{reject} \ H_t \mid H_t = \mathrm{true}). \tag{17}$$

The FDR is defined as the expectation of the ratio of the number of incorrectly rejected hypothesis (false discoveries) and that of all rejected hypothesis (discoveries):

$$\mathrm{FDR} = \mathbb{E}\left[\frac{IR}{R}\right], \quad (\mathrm{FDR} = 0 \quad \text{for } R = 0.) \tag{18}$$

To control the FDR at level $\alpha$, we proceed in the following way. Let $\{T(t)\}$ be ordered such that $T(1) \geq T(2) \cdots \geq T(N)$ and let the significance levels be given as:

$$\gamma_t = \alpha \frac{t}{N\eta}, \ t = 1, \cdots, N, \tag{19}$$

where $\eta$ is the harmonic number:

$$\eta = 1 + \frac{1}{2} + \cdots + \frac{1}{N} \approx \log N + 0.577. \tag{20}$$

Also let $\eta_t$ be the following quantile of the distribution of $\{T(t)\}$:

$$\mathrm{prob}(T(t) \geq \eta_t \mid H_t) = \gamma_t. \tag{21}$$

Finally let:

$$\widehat{t} = \max[t \mid T(t) \geq \eta_t]. \tag{22}$$

FDR rejects the $\widehat{t}$ hypotheses $(H_1, \cdots, H_{\widehat{t}})$ and accepts the remaining hypotheses $(H_{\widehat{t}+1}, \cdots, H_N)$. If the test statistics are known to be independent or positively correlated, then the following larger significance levels can be used:

$$\gamma_t = \alpha \frac{t}{N}, \ t = 1, \cdots, N. \tag{23}$$

## 4 Proposed Algorithm: RobGMM

In this section, we present an iterative procedure for solving (11). For the reader's convenience, we restate the optimization problem in (11):

$$\underset{\{N_i\}, \{\boldsymbol{\theta}_k\}, \{\mathbf{b}(t)\}}{\text{minimize}} \quad l\left(\boldsymbol{\theta}; \mathcal{D}, \{\mathbf{b}(t)\}\right) + \sum_{i=1}^{p} \sum_{t=1}^{N_i} \eta_t,$$

$$\text{subject to} \quad \boldsymbol{\pi}^T \mathbf{1} = 1, \quad \boldsymbol{\pi} \succeq 0, \quad \boldsymbol{\Sigma}_k \succ 0 \quad \forall k. \tag{24}$$

where $\eta_t$ are obtained by means of the FDR procedure, see (21) and (29) below. We minimize the above objective function alternatingly with respect to the outlier positions and the GMM parameters. First, we exploit the FDR-based penalty in (24) to arrive at an estimate of outlier positions given estimates of the GMM parameters. Then, for given cell outlier positions, we minimize (24) over the GMM parameters using the MM procedure. The MM-based derivation of the parameter updates avoids the introduction of latent variables and the computation of conditional expectations, as is typically required in alternative methods such as the EM algorithm.

### 4.1 Outlier Detection

In this subsection we describe the procedure for estimating the set $\{b_i(t)\}$. Our approach sequentially updates each binary variable, beginning with $\{b_1(t)\}_{t=1}^N$ while keeping the remaining variables $\{\hat{b}_j(t)\}_{j\neq 1}$ and the GMM parameters $\widehat{\boldsymbol{\theta}}$ fixed, followed by an analogous update for $\{b_2(t)\}_{t=1}^N$, and so on. To facilitate this, we first consider the negative log-likelihood of the observation $\mathbf{y}(t)$ given in (10) as a function of $\{b_i(t)\}$, using the latest estimates of $\{\hat{b}_j(t)\}_{j\neq i}$ and $\widehat{\boldsymbol{\theta}}$:

$$f_i\left(b_i(t)\right) \triangleq l_t\left(\widehat{\boldsymbol{\theta}}, \mathbf{y}(t), \{\hat{b}_j(t)\}_{j\neq i}, b_i(t)\right). \tag{25}$$

Since $b_i(t) \in \{0, 1\}$, the function in (25) can be expressed in an affine form as

$$f_i\left(b_i(t)\right) \triangleq b_i(t) f_i^1(t) + (1 - b_i(t)) f_i^0(t), \tag{26}$$

where $f_i^1(t)$ and $f_i^0(t)$ denote the values of $f_i\left(b_i(t)\right)$ at $b_i(t) = 1$ and $b_i(t) = 0$, respectively. Employing the representation in (26), the negative log-likelihood associated with the uncorrupted data can be written up to an additive constant as

$$\sum_{t=1}^{N} b_i(t) T_i(t), \tag{27}$$

where:

$$T_i(t) = f_i^1(t) - f_i^0(t), \quad t = 1, \cdots, N.$$

Each $T_i(t)$ can be viewed as a test statistic associated with the null hypothesis:

$\text{H}_i(t)$: The $i^{\text{th}}$ cell of $t^{\text{th}}$ sample belong to one of the mixture components and therefore is not an outlier.

Thus multiple hypotheses testing using FDR can be used to test these hypotheses jointly. Since the data samples are assumed to be independent (over $t$), the test statistics associated with these hypotheses are also independent and therefore the significance levels are given by (23). Let the thresholds $\eta_t$ be defined as in (21):

$$\text{prob}\left(T_i(t) \geq \eta_t \mid \text{H}_i(t)\right) = \gamma_t \tag{28}$$

By the log-likelihood ratio theorem (Stoica & Babu, 2024b) the variable $T_i(t)$ follows a chi-square distributions with one degree of freedom under the null hypothesis that $b_i(t) = 1$. Using this observation in (28) leads to an explicit choice of the penalty factor $\eta_t$ in (11):

$$\text{prob}\left(T_i(t) \geq \eta_t \mid T_i(t) \sim \chi^2(1)\right) = \frac{\alpha t}{N}, \qquad t = 1, \cdots, N, \tag{29}$$

where $\alpha$ denotes a predetermined false discovery rate, which is typically set at 0.05. Next observe that minimizing the likelihood in (11) with respect to $\{b_i(t)\}$ and $N_i$ is equivalent to solving the following minimization problem:

$$\min_{N_i, \{b_i(t)\}_{t=1}^N} \sum_{t=1}^{N} b_i(t) T_i(t) + \sum_{t=1}^{N_i} \eta_t. \tag{30}$$

By sorting the test statistics $\{T_i(t)\}_{t=1}^N$ in descending order, i.e., $T_i(1) \geq T_i(2) \geq \cdots \geq T_i(N)$ and minimizing (30) with respect to $\{b_i(t)\}_{t=1}^N$, the minimum of (30) with respect to $N_i$ is obtained by evaluating:

$$\sum_{t=N_i+1}^N T_i(t) + \sum_{t=1}^{N_i} \eta_t \tag{31}$$

for all possible $N_i$'s. Choosing the minimum of (31) yields the optimal number of corrupted cells, denoted by $\widehat{N_i}$, and consequently the corresponding estimates $\{\widehat{b}_i(t)\}_{t=1}^N$. The updates of the remaining variables $\{b_j(t)\}_{j\neq i}$ are derived similarly.

## 4.2 GMM parameter estimation

Using the latest estimates of the cell outlier positions $\{\widehat{\mathbf{b}}(t)\}$, the negative log-likelihood in (9) becomes:

$$l\left(\boldsymbol{\theta}; \mathcal{D}, \{\widehat{\mathbf{b}}(t)\}\right) = -\sum_{t=1}^N \log\left(\sum_{k=1}^K e^{g_{tk}\left(\boldsymbol{\theta}_k, \widehat{\mathbf{b}}(t)\right)}\right), \tag{32}$$

We will employ an MM algorithm to minimize (32) over $\boldsymbol{\theta}$. The objective in (32) is a log-sum-exp function (Boyd & Vandenberghe, 2004), therefore using Lemma 1 in Appendix A leads to the following tight upperbound at $\boldsymbol{\theta} = \widetilde{\boldsymbol{\theta}}$ (the latest estimate of $\boldsymbol{\theta}$):

$$l\left(\boldsymbol{\theta}; \mathcal{D}, \{\widehat{\mathbf{b}}(t)\}\right) \leq -\sum_{t=1}^N \sum_{k=1}^K w_{tk} g_{tk}\left(\boldsymbol{\theta}_k, \widehat{\mathbf{b}}(t)\right) + \text{constant}$$
$$\triangleq s_l\left(\boldsymbol{\theta} \mid \widetilde{\boldsymbol{\theta}}\right) + \text{constant}, \tag{33}$$

where

$$w_{tk} = \frac{e^{g_{tk}\left(\widetilde{\boldsymbol{\theta}}_k, \widehat{\mathbf{b}}(t)\right)}}{\sum_{j=1}^K e^{g_{tj}\left(\widetilde{\boldsymbol{\theta}}_j, \widehat{\mathbf{b}}(t)\right)}}. \tag{34}$$

We minimize the surrogate function $s_l\left(\boldsymbol{\theta} \mid \widetilde{\boldsymbol{\theta}}\right)$ to obtain the updated estimate $\widehat{\boldsymbol{\theta}}$:

$$\widehat{\boldsymbol{\theta}} = \arg\minimize_{\{\pi_k, \boldsymbol{\mu}_k, \boldsymbol{\Sigma}_k\}} \quad s_l\left(\boldsymbol{\theta} \mid \widetilde{\boldsymbol{\theta}}\right) \\ \text{subject to} \quad \boldsymbol{\pi}^\top \mathbf{1} = 1, \boldsymbol{\pi} \succeq 0, \boldsymbol{\Sigma}_k \succ 0 \,\forall k \tag{35}$$

Using (6), $s_l\left(\boldsymbol{\theta} \mid \widetilde{\boldsymbol{\theta}}\right)$ can be written (to with an additive constant) as

$$s_l\left(\boldsymbol{\theta} \mid \widetilde{\boldsymbol{\theta}}\right) = \sum_{t=1}^N \sum_{k=1}^K w_{tk}\left(\frac{1}{2}(\mathbf{y}(t) - \boldsymbol{\mu}_k)^\top \widehat{\mathbf{B}}_t \left(\widehat{\mathbf{B}}_t^\top \boldsymbol{\Sigma}_k \widehat{\mathbf{B}}_t\right)^{-1} \widehat{\mathbf{B}}_t^\top (\mathbf{y}(t) - \boldsymbol{\mu}_k) + \frac{1}{2}\log\left|\widehat{\mathbf{B}}_t^\top \boldsymbol{\Sigma}_k \widehat{\mathbf{B}}_t\right| - \log \pi_k\right). \tag{36}$$

Observe that $s_l\left(\boldsymbol{\theta} \mid \widetilde{\boldsymbol{\theta}}\right)$ is separable in $\{\pi_k\}$ and $\{\boldsymbol{\mu}_k, \boldsymbol{\Sigma}_k\}$, therefore the objective in (35) can be minimized separately with respect to $\{\pi_k\}$ and $\{\boldsymbol{\mu}_k, \boldsymbol{\Sigma}_k\}$.

### 4.2.1 Estimation of $\{\pi_k\}$

From (35), the minimization problem with respect to $\{\pi_k\}$ is given by:

$$
\begin{aligned}
\underset{\{\pi_k\}}{\text{maximize}} \quad & \sum_{t=1}^{N} \sum_{k=1}^{K} w_{tk} \log \pi_k \\
\text{subject to} \quad & \boldsymbol{\pi}^\top \mathbf{1} = 1, \boldsymbol{\pi} \succeq 0
\end{aligned}
\tag{37}
$$

Letting $\lambda$ be the multiplier for the equality constraint $\sum_{k=1}^{K} \pi_k = 1$, the Lagrangian for (37) is:

$$
\mathcal{L}(\{\pi_k\}, \lambda) = \sum_{t=1}^{N} \sum_{k=1}^{K} w_{tk} \log \pi_k + \lambda \left( \sum_{k=1}^{K} \pi_k - 1 \right).
\tag{38}
$$

The update of $\pi_k$ is obtained by setting the derivative of the Lagrangian with respect to $\pi_k$ to zero:

$$
\frac{\partial \mathcal{L}(\{\pi_k\}, \lambda)}{\partial \pi_k} = \frac{\sum_{t=1}^{N} w_{tk}}{\pi_k} + \lambda = 0.
\tag{39}
$$

The constraint $\sum_{k=1}^{K} \pi_k = 1$ and (39) implies that:

$$
\lambda = -\sum_{t=1}^{N} \sum_{k=1}^{K} w_{tk}.
\tag{40}
$$

By substituting $\lambda$ from (40) into (39), we get the update of $\{\pi_k\}$ as:

$$
\widehat{\pi}_k = \frac{\sum_{t=1}^{N} w_{tk}}{\sum_{t=1}^{N} \sum_{k=1}^{K} w_{tk}} \quad \forall k.
\tag{41}
$$

From (34) we get $\sum_{t=1}^{N} \sum_{k=1}^{K} w_{tk} = N$, therefore:

$$
\widehat{\pi}_k = \frac{\sum_{t=1}^{N} w_{tk}}{N} \quad \forall k.
\tag{42}
$$

### 4.2.2 Estimation of $\{\boldsymbol{\mu}_k\}$

Next consider the optimization problem with respect to $\{\boldsymbol{\mu}_k\}$ for fixed $\{\widehat{\mathbf{B}}_t\}$, $\{\widehat{\pi}_k\}$, and $\{\widehat{\boldsymbol{\Sigma}}_k\}$:

$$
\underset{\{\boldsymbol{\mu}_k\}}{\text{minimize}} \sum_{t=1}^{N} \sum_{k=1}^{K} w_{tk} \left( \mathbf{y}(t) - \boldsymbol{\mu}_k \right)^\top \widehat{\mathbf{B}}_t \left( \widehat{\mathbf{B}}_t^\top \widehat{\boldsymbol{\Sigma}}_k \widehat{\mathbf{B}}_t \right)^{-1} \widehat{\mathbf{B}}_t^\top \left( \mathbf{y}(t) - \boldsymbol{\mu}_k \right).
\tag{43}
$$

The solution to (43), which is a quadratic least-squares problem, is given by:

$$
\widehat{\boldsymbol{\mu}}_k = \left( \sum_{t=1}^{N} w_{tk} \widehat{\mathbf{B}}_t \left( \widehat{\mathbf{B}}_t^\top \widehat{\boldsymbol{\Sigma}}_k \widehat{\mathbf{B}}_t \right)^{-1} \widehat{\mathbf{B}}_t^\top \right)^{-1} \left( \sum_{t=1}^{N} w_{tk} \widehat{\mathbf{B}}_t \left( \widehat{\mathbf{B}}_t^\top \widehat{\boldsymbol{\Sigma}}_k \widehat{\mathbf{B}}_t \right)^{-1} \widehat{\mathbf{B}}_t^\top \mathbf{y}(t) \right) \quad \forall k.
\tag{44}
$$

### 4.2.3 Estimation of $\{\boldsymbol{\Sigma}_k\}$

With given $\{\widehat{\boldsymbol{\mu}}_k\}$ and $\{\widehat{\mathbf{B}}_t\}$ the optimization problem with respect to $\boldsymbol{\Sigma}_k$ is:

$$
\underset{\{\boldsymbol{\Sigma}_k \succeq \mathbf{0}\}}{\text{minimize}} \sum_{t=1}^{N} \sum_{k=1}^{K} w_{tk} \left( \log \left| \widehat{\mathbf{B}}_t^\top \boldsymbol{\Sigma}_k \widehat{\mathbf{B}}_t \right| + \mathbf{q}_k^\top(t) \widehat{\mathbf{B}}_t \left( \widehat{\mathbf{B}}_t^\top \boldsymbol{\Sigma}_k \widehat{\mathbf{B}}_t \right)^{-1} \widehat{\mathbf{B}}_t^\top \mathbf{q}_k(t) \right),
\tag{45}
$$

where $\mathbf{q}_k(t) = \mathbf{y}(t) - \widehat{\boldsymbol{\mu}}_k$. We tackle (45) using an one-step MM approach. An upper bound for the term $\log \left| \widehat{\mathbf{B}}_t^\top \boldsymbol{\Sigma}_k \widehat{\mathbf{B}}_t \right|$ in (45) for a given $\boldsymbol{\Sigma}_k = \widetilde{\boldsymbol{\Sigma}}_k$ can be obtained using Lemma 2 in Appendix A:

$$\sum_{t=1}^{N} w_{tk} \log |\widehat{\mathbf{B}}_t^\top \mathbf{\Sigma}_k \widehat{\mathbf{B}}_t| \leq \mathrm{Tr}\left(\mathbf{C}_k \mathbf{\Sigma}_k\right) + \mathrm{constant}, \tag{46}$$

where

$$\mathbf{C}_k \triangleq \sum_{t=1}^{N} w_{tk} \widehat{\mathbf{B}}_t (\widehat{\mathbf{B}}_t^\top \widetilde{\mathbf{\Sigma}}_k \widehat{\mathbf{B}}_t)^{-1} \widehat{\mathbf{B}}_t^\top. \tag{47}$$

Since both $\mathbf{\Sigma}_k$ and $\widetilde{\mathbf{\Sigma}}_k$ are positive semi-definite matrices, we can derive the following tight upperbound for the second term in (45) at $\widetilde{\mathbf{\Sigma}}_k$ using Lemma 3 in Appendix A:

$$\sum_{t=1}^{N} w_{tk} \mathbf{q}_k^\top(t) \widehat{\mathbf{B}}_t (\widehat{\mathbf{B}}_t^\top \mathbf{\Sigma}_k \widehat{\mathbf{B}}_t)^{-1} \widehat{\mathbf{B}}_t^\top \mathbf{q}_k(t) \leq \mathrm{Tr}\left(\mathbf{D}_k \mathbf{\Sigma}_k^{-1}\right), \tag{48}$$

where

$$\mathbf{D}_k \triangleq \sum_{t=1}^{N} w_{tk} \widetilde{\mathbf{\Sigma}}_k \widehat{\mathbf{B}}_t (\widehat{\mathbf{B}}_t^\top \widetilde{\mathbf{\Sigma}}_k \widehat{\mathbf{B}}_t)^{-1} \widehat{\mathbf{B}}_t^\top \mathbf{q}_k(t) \mathbf{q}_k^\top(t) \widehat{\mathbf{B}}_t (\widehat{\mathbf{B}}_t^\top \widetilde{\mathbf{\Sigma}}_k \widehat{\mathbf{B}}_t)^{-1} \widehat{\mathbf{B}}_t^\top \widetilde{\mathbf{\Sigma}}_k. \tag{49}$$

By combining (46) and (48), we arrive at the following surrogate minimization problem:

$$\min_{\{\mathbf{\Sigma}_k \succeq \mathbf{0}\}} \sum_{k=1}^{K} \left(\mathrm{Tr}\left(\mathbf{C}_k \mathbf{\Sigma}_k\right) + \mathrm{Tr}\left(\mathbf{D}_k \mathbf{\Sigma}_k^{-1}\right)\right). \tag{50}$$

Letting $\widetilde{\mathbf{E}}_k = \mathbf{D}_k^{-\frac{1}{2}} \mathbf{\Sigma}_k \mathbf{D}_k^{-\frac{1}{2}}$, the above problem can be equivalently written as:

$$\min_{\{\mathbf{\Sigma}_k \succeq \mathbf{0}\}} \sum_{k=1}^{K} \left(\mathrm{Tr}\left(\mathbf{D}_k^{\frac{1}{2}} \mathbf{C}_k \mathbf{D}_k^{\frac{1}{2}} \widetilde{\mathbf{E}}_k\right) + \mathrm{Tr}\left(\widetilde{\mathbf{E}}_k^{-1}\right)\right). \tag{51}$$

The matrix inequality

$$\left(\left(\mathbf{D}_k^{\frac{1}{2}} \mathbf{C}_k \mathbf{D}_k^{\frac{1}{2}}\right)^{\frac{1}{2}} - \widetilde{\mathbf{E}}_k^{-1}\right) \widetilde{\mathbf{E}}_k \left(\left(\mathbf{D}_k^{\frac{1}{2}} \mathbf{C}_k \mathbf{D}_k^{\frac{1}{2}}\right)^{\frac{1}{2}} - \widetilde{\mathbf{E}}_k^{-1}\right) \succeq \mathbf{0} \tag{52}$$

implies that

$$\mathrm{Tr}\left(\left(\mathbf{D}_k^{\frac{1}{2}} \mathbf{C}_k \mathbf{D}_k^{\frac{1}{2}}\right) \widetilde{\mathbf{E}}_k\right) + \mathrm{Tr}\left(\widetilde{\mathbf{E}}_k^{-1}\right) \geq 2\mathrm{Tr}\left(\left(\mathbf{D}_k^{\frac{1}{2}} \mathbf{C}_k \mathbf{D}_k^{\frac{1}{2}}\right)^{\frac{1}{2}}\right). \tag{53}$$

The minimum of left hand side of (53) is obtained when (see (52)):

$$\widetilde{\mathbf{E}}_k = \left(\mathbf{D}_k^{\frac{1}{2}} \mathbf{C}_k \mathbf{D}_k^{\frac{1}{2}}\right)^{-\frac{1}{2}}, \tag{54}$$

and thus corresponding minimizer of (50) is:

$$\widehat{\mathbf{\Sigma}}_k = \mathbf{D}_k^{\frac{1}{2}} \left(\mathbf{D}_k^{\frac{1}{2}} \mathbf{C}_k \mathbf{D}_k^{\frac{1}{2}}\right)^{-\frac{1}{2}} \mathbf{D}_k^{\frac{1}{2}}. \tag{55}$$

The pseudocode of the proposed Robust GMM (RobGMM) algorithm is summarized in Algorithm 1.

---

**Algorithm 1** RobGMM

---

**Input:** $\{\mathbf{y}(t)\}, \alpha = 0.05$
**Output:** $\{\widehat{b}_i(t)\}, \{\widehat{\pi}_k\}, \{\widehat{\boldsymbol{\mu}}_k\}$, and $\{\widehat{\boldsymbol{\Sigma}}_k\}$
 1: Initial values of $\{\pi_k\}, \{\boldsymbol{\mu}_k\}$, and $\{\boldsymbol{\Sigma}_k\}$ are obtained using a suitable clustering technique (such as K-means)
 2: $\{b_i(t) = 1\}$
 3: **repeat**
 4:     Obtain $\{\widehat{b}_i(t)\}$ using (30) and (31)
 5:     Obtain $\{\widehat{\pi}_k\}$ using (42)
 6:     Obtain $\{\widehat{\boldsymbol{\mu}}_k\}$ using (44)
 7:     Obtain $\{\widehat{\boldsymbol{\Sigma}}_k\}$ using (55)
 8: **until** Convergence

---

### 4.3 Convergence and Computational Complexity

The cyclic minimization of the objective in (11) with respect to $\{\pi_k, \mu_k, \boldsymbol{\Sigma}_k\}$, and $\{\mathbf{B}_t\}_{t=1}^N$ guarantees the monotonic decrease of the objective with respect to the iterative steps. Consequently the convergence of the objective function evaluated at the iterates of the RobGMM can be proved using techniques from Razaviyayn et al. (2013); Sun et al. (2016). The following convergence criterion for the RobGMM algorithm can be used:

$$\left| \frac{\mathcal{J}(\boldsymbol{\theta}^{\text{current}}, \{\mathbf{B}_t\}^{\text{current}}) - \mathcal{J}(\boldsymbol{\theta}^{\text{previous}}, \{\mathbf{B}_t\}^{\text{previous}})}{\mathcal{J}(\boldsymbol{\theta}^{\text{previous}}, \{\mathbf{B}_t\}^{\text{previous}})} \right| < \epsilon,$$

where $\mathcal{J}(\boldsymbol{\theta}^{\text{current}}, \{\mathbf{B}_t^{\text{current}}\})$ and $\mathcal{J}(\boldsymbol{\theta}^{\text{previous}}, \{\mathbf{B}_t^{\text{previous}}\})$ denote, respectively, the values of the penalized negative log-likelihood in (11) evaluated at the current and previous estimates of $\boldsymbol{\theta}$ and $\mathbf{B}_t$. In the simulation studies we have used $\epsilon = 10^{-6}$.

In the following, we analyze the per-iteration computational complexity of the proposed algorithm in terms of the number of mixture components $K$, data points $N$, and data dimension $p$. The dominant cost arises from evaluating the set of binary selection vectors $\{\mathbf{b}(t)\}$, which initially incurs $\mathcal{O}(KNp^4)$ operations. The computation of the mixing coefficients $\{\pi_k\}$ requires $\mathcal{O}(KNp^3 + K^2N)$ operations, the estimation of the means $\{\boldsymbol{\mu}_k\}$ requires $\mathcal{O}(Kp^3 + KNp^2 + KN)$, and the estimation of the covariance matrices $\{\boldsymbol{\Sigma}_k\}$ adds a further $\mathcal{O}(KNp^3 + KN)$ cost. Hence, the overall per-iteration complexity of the RobGMM algorithm is $\mathcal{O}(KNp^4 + K^2N)$.

The quartic dependence on $p$ primarily originates from the outlier detection stage, where the likelihood terms involve repeated inversions of covariance submatrices of the form $(\mathbf{B}_t^\top \widehat{\boldsymbol{\Sigma}}_k \mathbf{B}_t)^{-1}$, which costs $\mathcal{O}(p^3)$. However, during outlier detection, the component covariances $\{\boldsymbol{\Sigma}_k\}$ remain fixed and $\mathbf{B}_t$ changes only incrementally (typically by a single column addition or removal). This structure enables efficient updates of $(\mathbf{B}_t^\top \widehat{\boldsymbol{\Sigma}}_k \mathbf{B}_t)^{-1}$ using rank-one update and downdate formulas derived from the Sherman–Morrison–Woodbury identity and the Schur complement (block matrix inversion lemma) (Sherman & Morrison, 1950; Woodbury, 1950; Harville, 1997; Zhang, 2005; Stoica et al., 2005).

Let the current inverse be

$$\mathbf{H} = \left( (\boldsymbol{\Sigma}_k)_{\mathcal{S}, \mathcal{S}} \right)^{-1}, \tag{56}$$

where $\mathcal{S}$ is the index set corresponding to the uncorrupted dimensions. When a new index $i$ is added $(\mathcal{S}' = \mathcal{S} \cup \{i\})$, the inverse of the enlarged submatrix can be updated as

$$\left( (\boldsymbol{\Sigma}_k)_{\mathcal{S}', \mathcal{S}'} \right)^{-1} = \begin{bmatrix} \mathbf{H} + \mathbf{H}\boldsymbol{\beta}\,\alpha^{-1}\boldsymbol{\beta}^\top\mathbf{H} & -\mathbf{H}\boldsymbol{\beta}\,\alpha^{-1} \\ -\alpha^{-1}\boldsymbol{\beta}^\top\mathbf{H} & \alpha^{-1} \end{bmatrix}, \tag{57}$$

where $\boldsymbol{\beta} = (\boldsymbol{\Sigma}_k)_{\mathcal{S}, i}$ and $\alpha = (\boldsymbol{\Sigma}_k)_{i,i} - \boldsymbol{\beta}^\top\mathbf{H}\boldsymbol{\beta}$. Conversely, when an index is removed $(\mathcal{S} = \mathcal{S}' \setminus \{i\})$, and the previous inverse is partitioned as

$$\mathbf{H}' = \begin{bmatrix} \mathbf{F} & \boldsymbol{\gamma} \\ \boldsymbol{\gamma}^\top & \omega \end{bmatrix}, \tag{58}$$

the updated inverse corresponding to the reduced index set is obtained via the rank-one downdate

$$\left((\mathbf{\Sigma}_k)_{\mathcal{S},\mathcal{S}}\right)^{-1} = \mathbf{F} - \frac{1}{\omega}\,\boldsymbol{\gamma}\boldsymbol{\gamma}^{\top}. \tag{59}$$

Both updates in (57)–(59) involve matrix–vector multiplications and outer products, each costing $\mathcal{O}(p_t^2)$, where $p_t \leq p$. Hence, the per-update cost is upper-bounded by $\mathcal{O}(p^2)$, reducing the complexity of the outlier detection step from $\mathcal{O}(KNp^4)$ to $\mathcal{O}(KNp^3)$. Incorporating this improvement, the overall per-iteration computational complexity of the proposed RobGMM algorithm becomes $\mathcal{O}(KNp^3 + K^2N)$.

## 5 Numerical Study

In this section the performance of RobGMM for clustering as well as robust GMM parameter estimation is compared with that of two state-of-the-art robust methods, namely TCLUST (García-Escudero et al., 2008) and Student's $t$ mixture model (with degree of freedom = 5) (Peel & McLachlan, 2000). We have also included the vanila GMM (Dempster et al., 1977) in this comparison. Furthermore the RobGMM algorithm is also applied to the real-world *Top Gear* data set (Alfons, 2021) for clustering and outlier detection. Our experiments are run using the MATLAB software on Intel-i7 processor with 64 GB of RAM.

### 5.1 Clustering Performance

We evaluate the clustering performance of RobGMM algorithm on data generated using the simulation settings in Appendix B.1. We generate the outliers by randomly selecting 10% or 20% elements from the data matrix and modifying their values such that they are uniformly distributed in the interval $[-20, 20]$. Fig. 1 shows the uncorrupted data, the data corrupted with 10% cell outliers, and the clustering results obtained using competing methods and the proposed RobGMM algorithm. It can be seen that RobGMM algorithm has an excellent robust clustering performance.

The clustering performance of the RobGMM algorithm is compared with TCLUST and student's $t$ mixture using two performance metrics namely Accuracy (Metz, 1978) and the Enhanced Multivariate Pearson Correlation Coefficient (EMPC) (Stoica & Babu, 2024a). Evaluating these performance metrics requires computing the confusion matrix. Let $\mathbf{G}$ be the $(K+1) \times (K+1)$ confusion matrix in which the first $K$ groups are for the uncorrupted data clusters and the $(K+1)^{\text{th}}$ group is assigned to the outliers. The element $G_{i,j}$ of $\mathbf{G}$ shows the number of data points from the $i^{\text{th}}$ cluster that were classified as belonging to the $j^{\text{th}}$ cluster. Using $\mathbf{G}$ the Accuracy and EMPC are obtained as follows:

$$\text{Accuracy} = \frac{\sum_{k=1}^{K+1} G_{kk}}{\sum_{i=1}^{K+1}\sum_{j=1}^{K+1} G_{ij}}, \tag{60}$$

$$\text{EMPC} = \frac{1}{K+1}\sum_{k=1}^{K+1} \frac{(\delta_k + \beta_k)\,G_{kk}}{\delta_k \beta_k} - 1, \tag{61}$$

where $\delta_k = \sum_{j=1}^{K+1} G_{kj}$ and $\beta_k = \sum_{i=1}^{K+1} G_{ik}$. Note that to match the obtained clustering order to the true clustering order, we consider all possible orders of the $K$ clusters. For all these possible clustering orders, we construct the corresponding confusion matrices and compute the Accuracy for each of them. The clustering order that gives the maximum accuracy is identified as a match to the true clustering order.

We consider two cases: in case 1, the performance metrics are computed by considering 4 clusters and one outlier group, in this case the competing algorithms do not detect outliers. The average Accuracy and EMPC performance metrics (computed using 500 Monte-Carlo runs) for RobGMM and the competing methods in case 1 are shown in Table 1 and Table 2 for 10% and 20% outliers . In case 2, we compute the performance metrics after adding an outlier detection step to the competing algorithms. We first implement the competing methods on the corrupted data to estimate the distribution parameters (means, covariance matrices, and mixture proportions) and with these estimates we then run the FDR based multiple hypothesis testing to

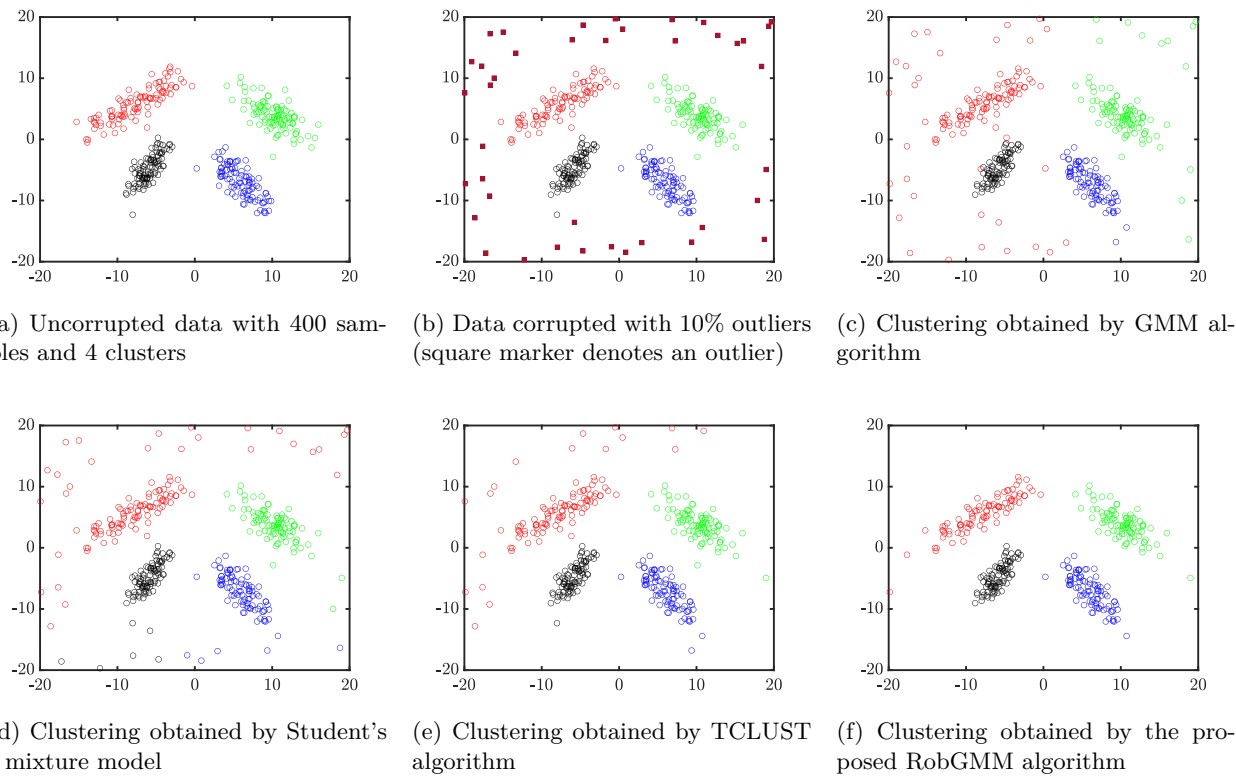

(a) Uncorrupted data with 400 samples and 4 clusters

(b) Data corrupted with 10% outliers (square marker denotes an outlier)

(c) Clustering obtained by GMM algorithm

(d) Clustering obtained by Student's $t$ mixture model

(e) Clustering obtained by TCLUST algorithm

(f) Clustering obtained by the proposed RobGMM algorithm

Figure 1: Uncorrupted and outlier-corrupted data, and clustering results obtained by competing methods and the proposed RobGMM algorithm.

detect the outliers for the competing methods. The obtained results for case 2 are shown in Table 3 and Table 4. From the results, we see that the RobGMM achieves better performance metrics in both the cases. The degradation in performance metrics for GMM, Student's $t$ mixture model, and TCLUST for case 2 when compared to case 1 is due to the fact that covariance matrices obtained via these methods are poorly estimated which leads to large misclassification of many inliers as outliers.

Table 1: Performance metrics for RobGMM and other methods for 10% outliers

|  | GMM | Student's $t$ mixture | TCLUST | RobGMM |
|---|---|---|---|---|
|  | (mean $\pm$ std. dev.) | (mean $\pm$ std. dev.) | (mean $\pm$ std. dev.) | (mean $\pm$ std. dev.) |
| **Accuracy** | $0.708 \pm 0.100$ | $0.821 \pm 0.106$ | $0.900 \pm 0.048$ | $0.981 \pm 0.008$ |
| **EMPC** | $0.378 \pm 0.195$ | $0.579 \pm 0.204$ | $0.714 \pm 0.099$ | $0.949 \pm 0.022$ |

Table 2: Performance metrics for RobGMM and other methods for 20% outliers

|  | GMM | Student's $t$ mixture | TCLUST | RobGMM |
|---|---|---|---|---|
|  | (mean $\pm$ std. dev.) | (mean $\pm$ std. dev.) | (mean $\pm$ std. dev.) | (mean $\pm$ std. dev.) |
| **Accuracy** | $0.580 \pm 0.094$ | $0.682 \pm 0.079$ | $0.718 \pm 0.100$ | $0.962 \pm 0.008$ |
| **EMPC** | $0.217 \pm 0.215$ | $0.426 \pm 0.169$ | $0.467 \pm 0.211$ | $0.947 \pm 0.015$ |

Table 3: Performance metrics for RobGMM and other methods for 10% outliers

|  | GMM | Student's $t$ mixture | TCLUST | RobGMM |
|---|---|---|---|---|
|  | (mean $\pm$ std. dev.) | (mean $\pm$ std. dev.) | (mean $\pm$ std. dev.) | (mean $\pm$ std. dev.) |
| **Accuracy** | $0.694 \pm 0.110$ | $0.809 \pm 0.105$ | $0.819 \pm 0.106$ | $0.980 \pm 0.006$ |
| **EMPC** | $0.134 \pm 0.220$ | $0.344 \pm 0.211$ | $0.349 \pm 0.215$ | $0.950 \pm 0.017$ |

Table 4: Performance metrics for RobGMM and other methods for 20% outliers

|  | GMM | Student's $t$ mixture | TCLUST | RobGMM |
|---|---|---|---|---|
|  | (mean $\pm$ std. dev.) | (mean $\pm$ std. dev.) | (mean $\pm$ std. dev.) | (mean $\pm$ std. dev.) |
| **Accuracy** | $0.542 \pm 0.095$ | $0.681 \pm 0.093$ | $0.690 \pm 0.092$ | $0.972 \pm 0.012$ |
| **EMPC** | $-0.063 \pm 0.218$ | $0.226 \pm 0.200$ | $0.192 \pm 0.198$ | $0.948 \pm 0.022$ |

## 5.2 Parameter Estimation Performance

The performance of the RobGMM algorithm for the GMM parameter estimation is compared with that of the competing methods in different scenarios using the following metrics: normalized root mean squared error (NRMSE) and the Kullback-Leibler (KL) divergence. The NRMSE of the estimated mean and covariance matrix for the $k^{\text{th}}$ cluster are given by:

$$\text{NRMSE}\left(\widehat{\boldsymbol{\mu}}_k\right) = \frac{\|\boldsymbol{\mu}_k - \widehat{\boldsymbol{\mu}}_k\|}{\|\boldsymbol{\mu}_k\|}, \tag{62}$$

$$\text{NRMSE}\left(\widehat{\boldsymbol{\Sigma}}_k\right) = \frac{\|\boldsymbol{\Sigma}_k - \widehat{\boldsymbol{\Sigma}}_k\|_F}{\|\boldsymbol{\Sigma}_k\|_F}, \tag{63}$$

where $\|\cdot\|_F$ denotes the Frobenius norm. The KL divergence of the $k^{\text{th}}$ component is defined as:

$$\text{KL}\left[\left(\boldsymbol{\mu}_k, \boldsymbol{\Sigma}_k\right) || \left(\widehat{\boldsymbol{\mu}}_k, \widehat{\boldsymbol{\Sigma}}_k\right)\right] = \frac{1}{2}\left[\log\left|\widehat{\boldsymbol{\Sigma}}_k\right| - \log|\boldsymbol{\Sigma}_k| - p + \text{Tr}\left(\widehat{\boldsymbol{\Sigma}}_k^{-1}\boldsymbol{\Sigma}_k\right) + \left(\widehat{\boldsymbol{\mu}}_k - \boldsymbol{\mu}_k\right)^\top \widehat{\boldsymbol{\Sigma}}_k^{-1}\left(\widehat{\boldsymbol{\mu}}_k - \boldsymbol{\mu}_k\right)\right]. \tag{64}$$

The cellwise outliers are generated by randomly selecting cell indices from the data matrix and perturbing their values by random shifts drawn from a uniform distribution. To ensure that these artificially introduced

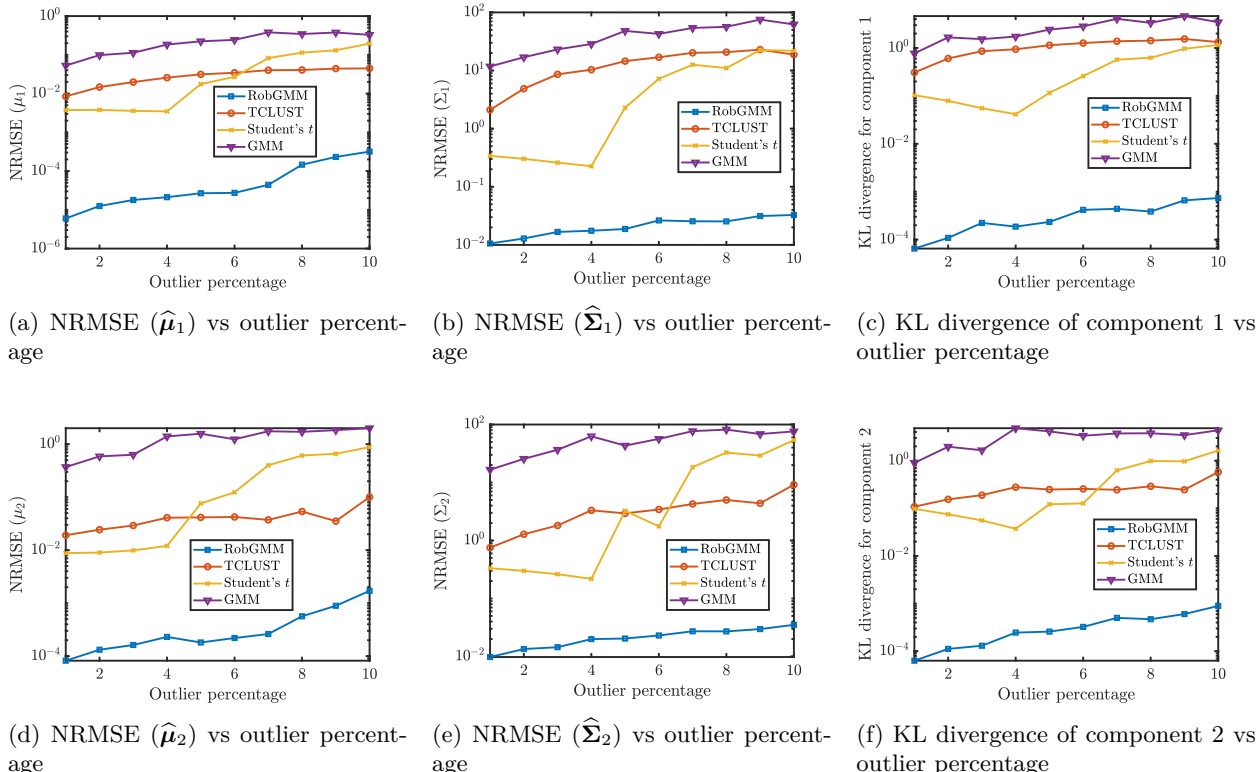

(a) NRMSE ($\widehat{\boldsymbol{\mu}}_1$) vs outlier percentage

(b) NRMSE ($\widehat{\boldsymbol{\Sigma}}_1$) vs outlier percentage

(c) KL divergence of component 1 vs outlier percentage

(d) NRMSE ($\widehat{\boldsymbol{\mu}}_2$) vs outlier percentage

(e) NRMSE ($\widehat{\boldsymbol{\Sigma}}_2$) vs outlier percentage

(f) KL divergence of component 2 vs outlier percentage

Figure 2: NRMSE and KL divergence for 2 clusters with $p = 2$.

outliers do not overlap with the genuine cluster regions, each perturbed value is checked to verify that its distance from all cluster means exceeds three standard deviations of the corresponding clusters. If this condition is not satisfied, the perturbation is re-sampled until the shifted values lie outside the cluster regions, that is, beyond the 99$^{\text{th}}$ percentile of any cluster component. It is important to note that this restriction applies only to the outlier generation procedure; the outlier detection in RobGMM operates independently based on the estimated model parameters. The simulations are performed for $K \in \{2, 3\}$ clusters with different choices of the number of dimensions ($p \in \{2, 5, 10, 25\}$). The simulation settings for these experiments are given in Appendix B.2. The obtained results for these simulations are shown in Figure 2, Figure 3, Figure 4, and Figure 5. From the plots of NRMSE and KL divergence versus outlier percentage, we observe that, compared to the competing methods, RobGMM exhibits superior parameter estimation performance across a practical range of data dimensions.

## 5.3 Top Gear Dataset Clustering

The *Top Gear* dataset comprises various car models and their numerical feature specifications. We selected $p = 11$ fully observed features corresponding to $N = 245$ car models. We used a logarithmic transformation of the highly skewed variables *price, displacement, brake horsepower (BHP), torque,* and *top speed* (Alfons, 2021). We then applied the RobGMM algorithm to the dataset. In Fig. 6, we show the penalized negative log-likelihood objective vs RobGMM iterations, and as expected the objective converges monotonically.

The car models were grouped into four clusters using the RobGMM algorithm: Cluster 1 comprises high-performance and luxury sports cars such as the *Aston Martin V12 Zagato, Ferrari 458, Audi R8, Aston Martin Vanquish, Aston Martin Vantage, Audi A5 Sportback, Audi A7 Sportback, Audi R8 V10, Bentley Continental GTC, BMW 6 Series, Corvette C6, Porsche 911, Porsche Boxster, Jaguar XFR,* and *Lamborghini Aventador.* These cars are distinguished by their emphasis on power, speed, and exclusivity. Cluster 2 consists largely of mainstream, compact, and family-oriented vehicles including models like *Ford Fiesta,*

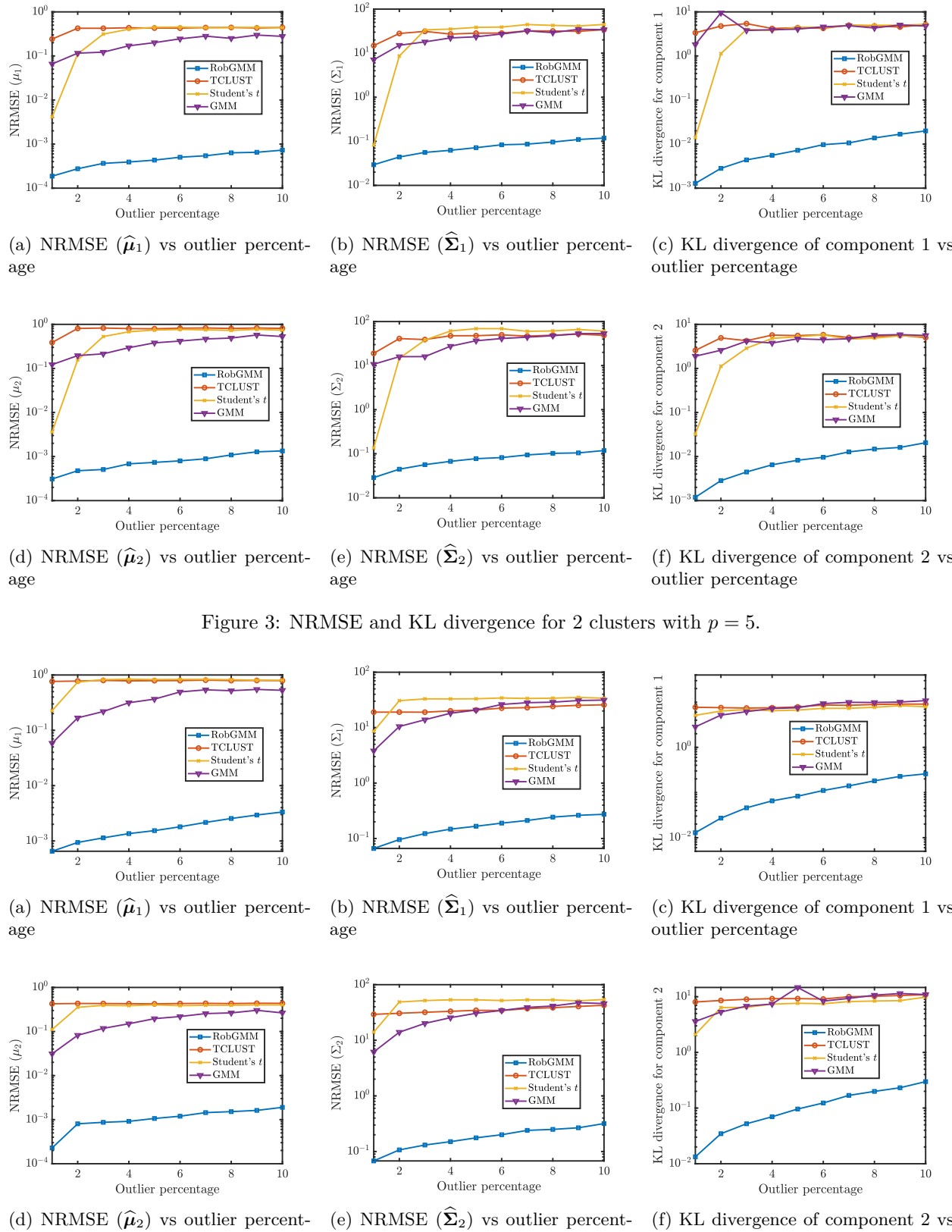

Figure 3: NRMSE and KL divergence for 2 clusters with $p = 5$.

(a) NRMSE $(\widehat{\boldsymbol{\mu}}_1)$ vs outlier percentage

(b) NRMSE $(\widehat{\boldsymbol{\Sigma}}_1)$ vs outlier percentage

(c) KL divergence of component 1 vs outlier percentage

(d) NRMSE $(\widehat{\boldsymbol{\mu}}_2)$ vs outlier percentage

(e) NRMSE $(\widehat{\boldsymbol{\Sigma}}_2)$ vs outlier percentage

(f) KL divergence of component 2 vs outlier percentage

Figure 4: NRMSE and KL divergence for 2 clusters with $p = 10$.

(a) NRMSE $(\widehat{\boldsymbol{\mu}}_1)$ vs outlier percentage

(b) NRMSE $(\widehat{\boldsymbol{\Sigma}}_1)$ vs outlier percentage

(c) KL divergence of component 1 vs outlier percentage

(d) NRMSE $(\widehat{\boldsymbol{\mu}}_2)$ vs outlier percentage

(e) NRMSE $(\widehat{\boldsymbol{\Sigma}}_2)$ vs outlier percentage

(f) KL divergence of component 2 vs outlier percentage

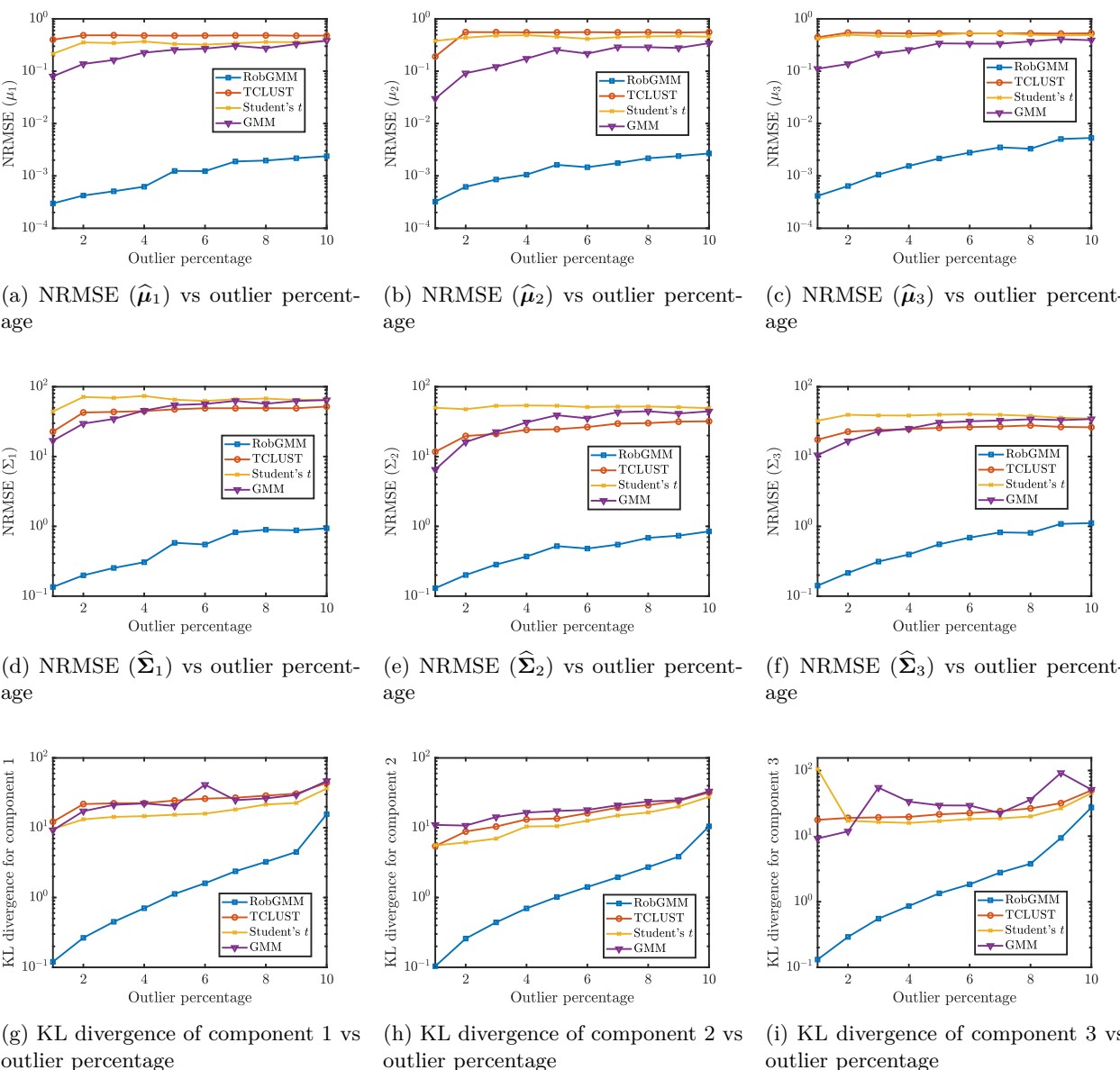

(a) NRMSE ($\widehat{\boldsymbol{\mu}}_1$) vs outlier percentage

(b) NRMSE ($\widehat{\boldsymbol{\mu}}_2$) vs outlier percentage

(c) NRMSE ($\widehat{\boldsymbol{\mu}}_3$) vs outlier percentage

(d) NRMSE ($\widehat{\boldsymbol{\Sigma}}_1$) vs outlier percentage

(e) NRMSE ($\widehat{\boldsymbol{\Sigma}}_2$) vs outlier percentage

(f) NRMSE ($\widehat{\boldsymbol{\Sigma}}_3$) vs outlier percentage

(g) KL divergence of component 1 vs outlier percentage

(h) KL divergence of component 2 vs outlier percentage

(i) KL divergence of component 3 vs outlier percentage

Figure 5: NRMSE and KL divergence for 3 clusters with $p = 25$.

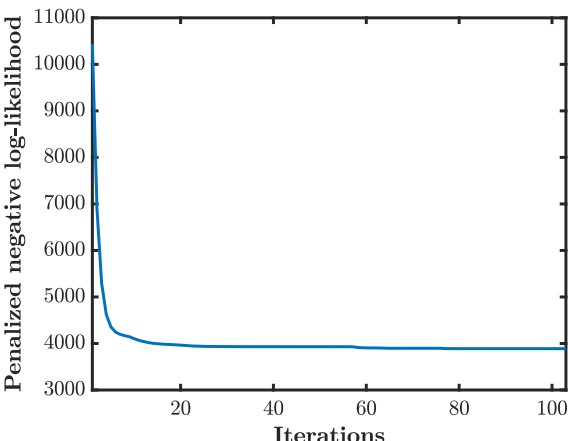

Figure 6: Penalized negative log-likelihood objective vs iterations of RobGMM on the *Top Gear* dataset

*Honda Civic, Alfa Romeo Giulietta, Aston Martin Cygnet, Volkswagen Tiguan, Peugeot 107, Renault Clio, Renault Megane, SEAT Toledo, Toyota iQ, Volkswagen Passat, Chevrolet Spark, Hyundai i20, Suzuki Swift,* and *Volkswagen Golf.* These cars are practical, widely available, and suited for everyday use. Cluster 3 includes large SUVs and executive luxury vehicles such as *Rolls-Royce Phantom, Bentley Mulsanne, Audi Q7, BMW X6, Land Rover Range Rover, Rolls-Royce Ghost, Rolls-Royce Phantom Coupe, Toyota Land Cruiser V8, Volkswagen Touareg, Bentley Flying Spur, Jeep Wrangler, Nissan Pathfinder, Ssangyong Rodius,* and *Lexus RX*, indicating a focus on comfort, size, and premium features. Cluster 4 is composed of hybrid, electric, and niche vehicles including *Audi TT Coupe, Audi TT Roadster, BMW i3, Chevrolet Volt, Fiat Doblo, Lexus CT 200h, Peugeot 207 CC, Peugeot 3008, Peugeot 308 CC, Subaru BRZ, Suzuki Grand Vitara, Toyota GT 86, Toyota Prius, Vauxhall Ampera,* and *Citroen DS5,* with a focus on eco-friendliness and innovation.

In Fig. 7, we show some examples of the cell outliers detected by RobGMM, which highlight feature values that deviate significantly from the typical characteristics of cars within their respective clusters (ten representative cars randomly selected from each cluster are shown in the figure). For instance, in Cluster 1, the *Bentley Continental GTC* is notably heavy for a performance sports car. In Cluster 2, the *Peugeot 107* is exceptionally lightweight for a mainstream everyday vehicle, while the *Aston Martin Cygnet* and *Toyota iQ* exhibit unusually short body lengths despite being categorized as regular city cars. In Cluster 3, the *Bentley Mulsanne*, *Rolls-Royce Ghost*, *Rolls-Royce Phantom*, and *Rolls-Royce Phantom Coupe* possess body lengths significantly greater than those of other luxury sedans and SUVs. In Cluster 4, the *BMW i3* records an exceptionally high MPG value.

In a final experiment, we inject *cellwise* contamination by randomly selecting 1% of cells (independently for each dimension) and perturbing only those entries, leaving all other cells of the data sample unchanged. We then apply all the methods to the contaminated data. For GMM, Student's $t$ mixture model, and TCLUST, we detect the cell-outlier positions using the FDR-based multiple hypothesis testing procedure applied to the distribution parameters (means, covariance matrices, and weights) estimated by these methods. The true and detected cell-outlier positions are shown in Figure 8. The results demonstrate that the proposed method detects the cell-outlier positions more accurately than the competing methods.

## 6 Conclusion

In this paper, we proposed a novel algorithm for robust estimation of GMM parameters and clustering that explicitly accounts for cell outliers. We minimized a penalized negative log-likelihood function to estimate the GMM parameters where the penalty term was derived from the FDR principle. The penalized negative log-likelihood function was cyclically minimized over the outlier positions and the GMM parameters. The minimization over the GMM parameters was done via an MM framework in which we minimized a tight

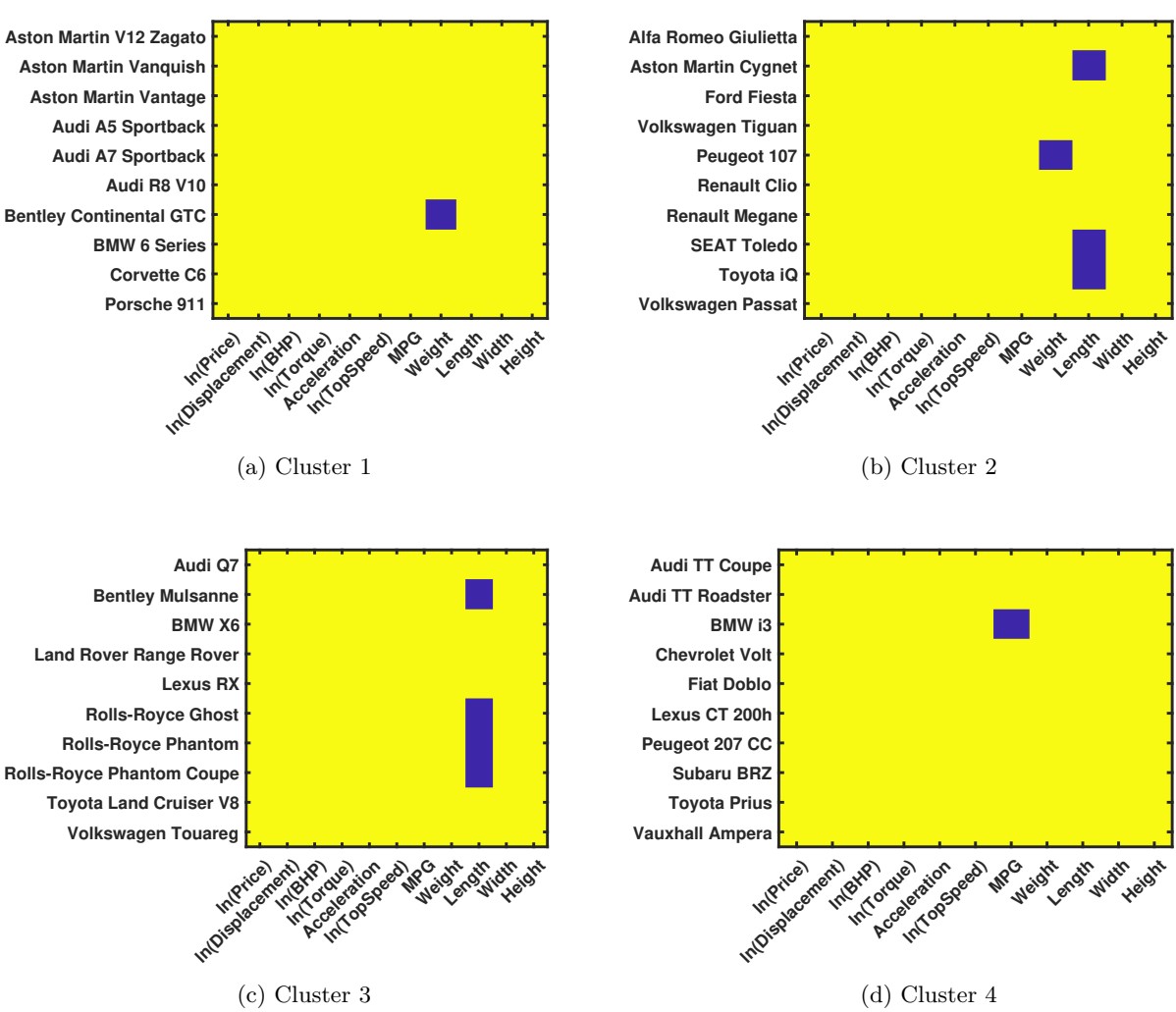

Figure 7: Cell outliers detected by RobGMM method in the *Top Gear* dataset (Violet pixel shows a cell outlier).

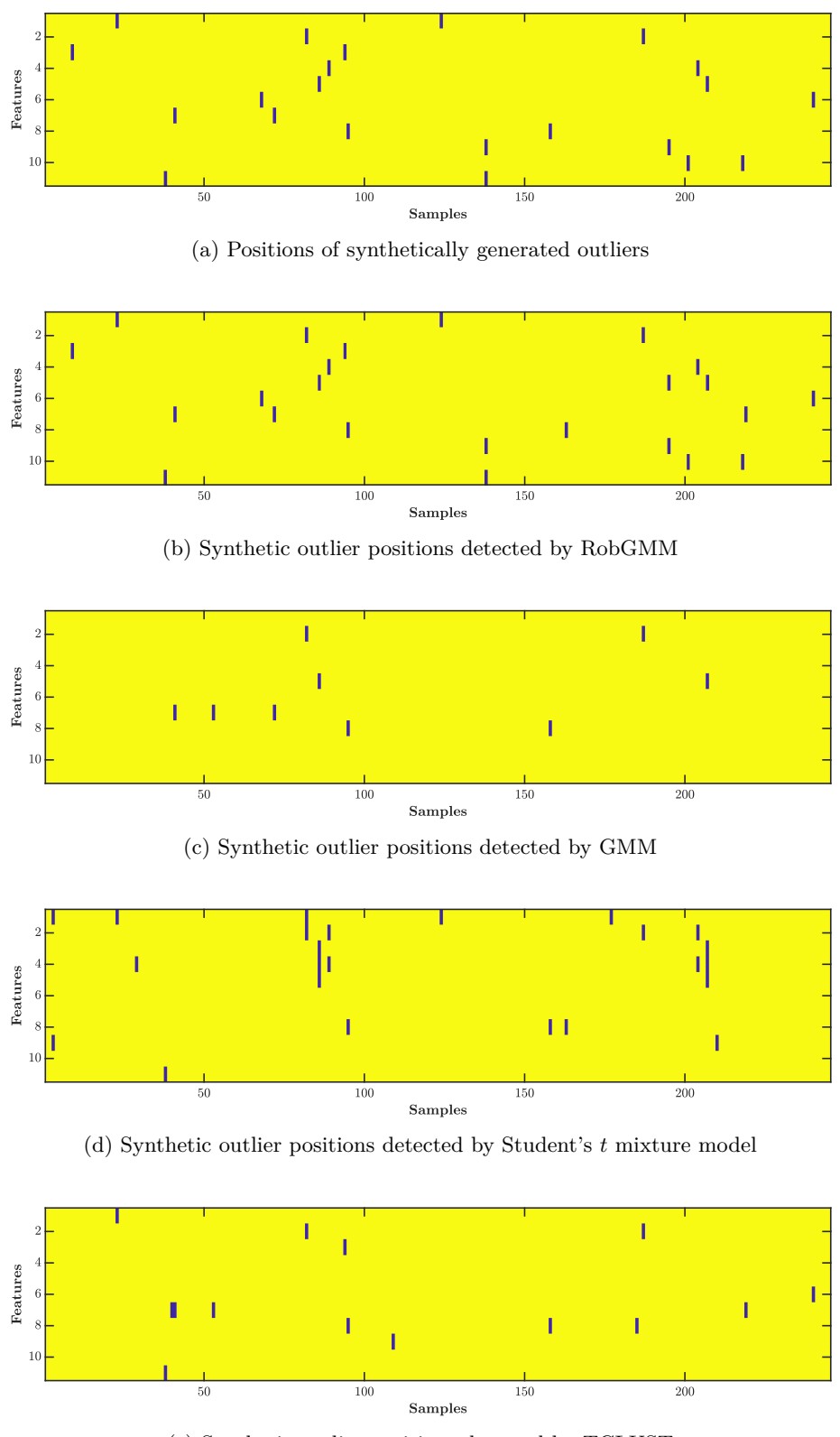

(a) Positions of synthetically generated outliers

(b) Synthetic outlier positions detected by RobGMM

(c) Synthetic outlier positions detected by GMM

(d) Synthetic outlier positions detected by Student's $t$ mixture model

(e) Synthetic outlier positions detected by TCLUST

Figure 8: Cell outliers detected by competing methods in the *Top Gear* dataset (Violet pixel shows a cell outlier).

upper bound on the negative log-likelihood function. This minimization operation decoupled into simpler optimization subproblems that were solved efficiently. The present study can be extended to non-Gaussian mixtures, such as the Student's $t$ mixture model, by deriving the corresponding penalized negative log-likelihood function, formulating the appropriate test statistics and their distributions under the Student's $t$ assumption, and subsequently estimating the distribution parameters using the MM framework. We presented an extensive simulation study, comprising various experiments aimed at evaluating the performance of the proposed method on synthetic as well as real-world data and at comparing it with state-of-the-art robust GMM parameter estimation and clustering techniques.

## Acknowledgment

This research was financially supported by the project *Blending probabilistic and nonlinear representations* (contract number: 2025-04318), funded by the Swedish Research Council.

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

## A   Supporting Lemmas and Their Proofs

**Lemma 1.** *The log-sum-exp function can be lower-bounded at any $\widetilde{\mathbf{x}}$ as follows:*

$$h\left(\mathbf{x}\right) \triangleq \log\left(\sum_{i=1}^{p} e^{x_i}\right) \geq h\left(\widetilde{\mathbf{x}}\right) + \nabla h\left(\widetilde{\mathbf{x}}\right)^\top \left(\mathbf{x} - \widetilde{\mathbf{x}}\right), \tag{65}$$

*with equality achieved at $\mathbf{x} = \widetilde{\mathbf{x}}$. Here*

$$\mathbf{x} = [x_1, \cdots, x_p]^\top$$

*and $\nabla h\left(\widetilde{\mathbf{x}}\right)$ denotes the gradient of $h\left(\mathbf{x}\right)$ at $\widetilde{\mathbf{x}}$. The gradient of $h\left(\mathbf{x}\right)$ is given by:*

$$\nabla h\left(\mathbf{x}\right) = \left(\sum_{i=1}^{p} e^{x_i}\right)^{-1} \begin{pmatrix} e^{x_1} \\ \vdots \\ e^{x_p} \end{pmatrix}. \tag{66}$$

*Proof.* The log-sum-exp function $h\left(\mathbf{x}\right)$ is convex on $\mathbb{R}^{p \times 1}$ (Boyd & Vandenberghe, 2004), therefore a tight lower bound for $h\left(\mathbf{x}\right)$ at any $\widetilde{\mathbf{x}}$ can be constructed by using the following first order Taylor expansion:

$$h\left(\mathbf{x}\right) \geq h\left(\widetilde{\mathbf{x}}\right) + \nabla h\left(\widetilde{\mathbf{x}}\right)^\top \left(\mathbf{x} - \widetilde{\mathbf{x}}\right). \tag{67}$$

$\square$

**Lemma 2.** *Let $\boldsymbol{\Sigma}_k$ and $\widetilde{\boldsymbol{\Sigma}}_k$ be positive semi-definite matrices. Then the following inequality holds for any matrix $\widehat{\mathbf{B}}_t$:*

$$\log\left|\widehat{\mathbf{B}}_t^\top \boldsymbol{\Sigma}_k \widehat{\mathbf{B}}_t\right| \leq \mathrm{Tr}\left(\left(\widehat{\mathbf{B}}_t^\top \widetilde{\boldsymbol{\Sigma}}_k \widehat{\mathbf{B}}_t\right)^{-1} \widehat{\mathbf{B}}_t^\top \boldsymbol{\Sigma}_k \widehat{\mathbf{B}}_t\right) + \text{constant}, \tag{68}$$

*with equality for $\boldsymbol{\Sigma}_k = \widetilde{\boldsymbol{\Sigma}}_k$.*

*Proof.* From the arithmetic mean–geometric mean inequality (Sun et al., 2016) we have that:

$$\left|\left(\widehat{\mathbf{B}}_t^\top \widetilde{\boldsymbol{\Sigma}}_k \widehat{\mathbf{B}}_t\right)^{-1} \widehat{\mathbf{B}}_t^\top \boldsymbol{\Sigma}_k \widehat{\mathbf{B}}_t\right|^{1/p} \leq \frac{1}{p}\mathrm{Tr}\left(\left(\widehat{\mathbf{B}}_t^\top \widetilde{\boldsymbol{\Sigma}}_k \widehat{\mathbf{B}}_t\right)^{-1} \widehat{\mathbf{B}}_t^\top \boldsymbol{\Sigma}_k \widehat{\mathbf{B}}_t\right) \tag{69}$$

or, equivalently,

$$\frac{1}{p}\left[\log\left|\widehat{\mathbf{B}}_t^\top \boldsymbol{\Sigma}_k \widehat{\mathbf{B}}_t\right| - \log\left|\widehat{\mathbf{B}}_t^\top \widetilde{\boldsymbol{\Sigma}}_k \widehat{\mathbf{B}}_t\right|\right] \leq \log\left[\frac{1}{p}\mathrm{Tr}\left(\left(\widehat{\mathbf{B}}_t^\top \widetilde{\boldsymbol{\Sigma}}_k \widehat{\mathbf{B}}_t\right)^{-1} \widehat{\mathbf{B}}_t^\top \boldsymbol{\Sigma}_k \widehat{\mathbf{B}}_t\right)\right] \triangleq \log(x) \tag{70}$$

Since the function $\log(x)$ is concave, it can be upper bounded by its first-order Taylor expansion at any point $x_0$:

$$\log(x) \leq \log(x_0) + \frac{1}{x_0}(x - x_0) \tag{71}$$

In particular, when $x_0 = 1$ this inequality simplifies to:

$$\log(x) \leq x - 1 \tag{72}$$

Combining (70) and (72) yields

$$\log \left| \widehat{\mathbf{B}}_t^\top \boldsymbol{\Sigma}_k \widehat{\mathbf{B}}_t \right| \leq \mathrm{Tr} \left( \left( \widehat{\mathbf{B}}_t^\top \widetilde{\boldsymbol{\Sigma}}_k \widehat{\mathbf{B}}_t \right)^{-1} \widehat{\mathbf{B}}_t^\top \boldsymbol{\Sigma}_k \widehat{\mathbf{B}}_t \right) + \text{constant}, \tag{73}$$

where $\text{constant} = \log \left| \widehat{\mathbf{B}}_t^\top \widetilde{\boldsymbol{\Sigma}}_k \widehat{\mathbf{B}}_t \right| - p$.

$\square$

**Lemma 3.** *Let $\boldsymbol{\Sigma}_k$ and $\widetilde{\boldsymbol{\Sigma}}_k$ be positive semi-definite matrices. Then the following inequality holds for any matrix $\widehat{\mathbf{B}}_t$:*

$$\left( \widehat{\mathbf{B}}_t^\top \boldsymbol{\Sigma}_k \widehat{\mathbf{B}}_t \right)^{-1} \preceq \left( \widehat{\mathbf{B}}_t^\top \widetilde{\boldsymbol{\Sigma}}_k \widehat{\mathbf{B}}_t \right)^{-1} \widehat{\mathbf{B}}_t^\top \widetilde{\boldsymbol{\Sigma}}_k \boldsymbol{\Sigma}_k^{-1} \widetilde{\boldsymbol{\Sigma}}_k \widehat{\mathbf{B}}_t \left( \widehat{\mathbf{B}}_t^\top \widetilde{\boldsymbol{\Sigma}}_k \widehat{\mathbf{B}}_t \right)^{-1} \tag{74}$$

*with equality for $\boldsymbol{\Sigma}_k = \widetilde{\boldsymbol{\Sigma}}_k$.*

*Proof.* Consider the matrix:

$$\begin{bmatrix} \widehat{\mathbf{B}}_t^\top \widetilde{\boldsymbol{\Sigma}}_k \boldsymbol{\Sigma}_k^{-1} \widetilde{\boldsymbol{\Sigma}}_k \widehat{\mathbf{B}}_t & \widehat{\mathbf{B}}_t^\top \widetilde{\boldsymbol{\Sigma}}_k \widehat{\mathbf{B}}_t \\ \widehat{\mathbf{B}}_t^\top \widetilde{\boldsymbol{\Sigma}}_k \widehat{\mathbf{B}}_t & \widehat{\mathbf{B}}_t^\top \boldsymbol{\Sigma}_k \widehat{\mathbf{B}}_t \end{bmatrix} = \begin{bmatrix} \widehat{\mathbf{B}}_t^\top \widetilde{\boldsymbol{\Sigma}}_k \boldsymbol{\Sigma}_k^{-1/2} \\ \widehat{\mathbf{B}}_t^\top \boldsymbol{\Sigma}_k^{1/2} \end{bmatrix} \begin{bmatrix} \boldsymbol{\Sigma}_k^{-1/2} \widetilde{\boldsymbol{\Sigma}}_k \widehat{\mathbf{B}}_t & \boldsymbol{\Sigma}_k^{1/2} \widehat{\mathbf{B}}_t \end{bmatrix} \succeq \mathbf{0}$$

Because this matrix is positive semidefinite, its Schur complements are also positive semidefinite. In particular:

$$\widehat{\mathbf{B}}_t^\top \widetilde{\boldsymbol{\Sigma}}_k \boldsymbol{\Sigma}_k^{-1} \widetilde{\boldsymbol{\Sigma}}_k \widehat{\mathbf{B}}_t \succeq \left( \widehat{\mathbf{B}}_t^\top \widetilde{\boldsymbol{\Sigma}}_k \widehat{\mathbf{B}}_t \right) \left( \widehat{\mathbf{B}}_t^\top \boldsymbol{\Sigma}_k \widehat{\mathbf{B}}_t \right)^{-1} \left( \widehat{\mathbf{B}}_t^\top \widetilde{\boldsymbol{\Sigma}}_k \widehat{\mathbf{B}}_t \right), \tag{75}$$

which is equivalent to:

$$\left( \widehat{\mathbf{B}}_t^\top \boldsymbol{\Sigma}_k \widehat{\mathbf{B}}_t \right)^{-1} \preceq \left( \widehat{\mathbf{B}}_t^\top \widetilde{\boldsymbol{\Sigma}}_k \widehat{\mathbf{B}}_t \right)^{-1} \widehat{\mathbf{B}}_t^\top \widetilde{\boldsymbol{\Sigma}}_k \boldsymbol{\Sigma}_k^{-1} \widetilde{\boldsymbol{\Sigma}}_k \widehat{\mathbf{B}}_t \left( \widehat{\mathbf{B}}_t^\top \widetilde{\boldsymbol{\Sigma}}_k \widehat{\mathbf{B}}_t \right)^{-1}. \tag{76}$$

$\square$

# B Simulation Settings

The detailed simulation settings used in the numerical experiments are provided below.

## B.1 For Clustering Performance

The simulation settings for evaluating the clustering performance are as follows:

- Samples $(N) = 400$

- Dimensions $(p) = 2$ (chosen small for the sake of displaying the clustering results)

- Clusters $(K) = 4$

- Proportions $(\boldsymbol{\pi}) = [0.25, 0.25, 0.25, 0.25]$

- Means: $\boldsymbol{\mu}_1 = \begin{bmatrix} -7 \\ 6 \end{bmatrix}$, $\boldsymbol{\mu}_2 = \begin{bmatrix} 6 \\ -7 \end{bmatrix}$, $\boldsymbol{\mu}_3 = \begin{bmatrix} 10 \\ 4 \end{bmatrix}$, $\boldsymbol{\mu}_4 = \begin{bmatrix} -6 \\ -5 \end{bmatrix}$

- Covariances: $\boldsymbol{\Sigma}_1 = \begin{bmatrix} 9.6 & 5.9 \\ 5.9 & 6.0 \end{bmatrix}$, $\boldsymbol{\Sigma}_2 = \begin{bmatrix} 3.6 & -3.0 \\ -3.0 & 5.9 \end{bmatrix}$, $\boldsymbol{\Sigma}_3 = \begin{bmatrix} 5.8 & -4.1 \\ -4.1 & 6.0 \end{bmatrix}$, $\boldsymbol{\Sigma}_4 = \begin{bmatrix} 1.6 & 2.2 \\ 2.2 & 4.9 \end{bmatrix}$

## B.2 For Parameter Estimation Performance

The experiments are performed with different choices of number of dimensions ($p \in \{2, 5, 10, 25\}$). The synthetic data is generated for $K \in \{2, 3\}$ clusters. Each cluster is modeled as a multivariate Gaussian distribution with distinct mean and covariance structures, constructed as follows.

- **Cluster Means:** For each feature dimension $d \in \{1, \ldots, p\}$, the mean values across $K$ clusters were assigned as a random permutation of a fixed set of base mean values, ensuring that the cluster centers remain well separated in every dimension:

$$\mathbf{M} = \begin{bmatrix} \boldsymbol{\mu}_1 & \boldsymbol{\mu}_2 & \cdots & \boldsymbol{\mu}_K \end{bmatrix} \in \mathbb{R}^{p \times K}, \qquad \mu_{dk} \in \begin{cases} \text{Perm}\{5, 25\}, & K = 2, \\ \text{Perm}\{5, 20, 35\}, & K = 3. \end{cases}$$

- **Cluster Covariances:** For each cluster $k$, a symmetric positive definite covariance matrix $\boldsymbol{\Sigma}_k \in \mathbb{R}^{p \times p}$ was generated as

$$\boldsymbol{\Sigma}_k = \mathbf{U}_k \boldsymbol{\Lambda}_k \mathbf{U}_k^\top,$$

where $\mathbf{U}_k$ denotes the eigenvector matrix obtained from the eigendecomposition of a random symmetric matrix

$$\mathbf{R}_k = \mathbf{A}_k \mathbf{A}_k^\top, \qquad [\mathbf{A}_k]_{i,j} \sim \mathcal{N}(0, 1).$$

The diagonal eigenvalue matrix $\boldsymbol{\Lambda}_k = \text{diag}(\lambda_{k1}, \ldots, \lambda_{kp})$ contains independent entries drawn from uniform distributions with cluster-specific spectral ranges:

$$\lambda_{ki} \sim \begin{cases} \mathcal{U}(6, 15), & k = 1, \\ \mathcal{U}(4, 10), & k = 2, \\ \mathcal{U}(3, 8), & k = 3 \text{ (if } K = 3). \end{cases}$$

This procedure yields full-rank, well-conditioned covariance matrices with diverse eigenvalue spectra, thereby producing clusters of distinct orientation and spread.

- **Sample Generation:** For each cluster $k$, data samples were generated as

$$\mathbf{X}_k = \boldsymbol{\mu}_k \mathbf{1}_{\pi_k N}^\top + \boldsymbol{\Sigma}_k^{1/2} \mathbf{Z}_k, \qquad \mathbf{Z}_k \in \mathbb{R}^{p \times \pi_k N}, \ \mathbf{z}_{k,t} \sim \mathcal{N}(\mathbf{0}, \mathbf{I}_p),$$

where each column $\mathbf{z}_{k,t}$ of $\mathbf{Z}_k$ represents an independent $p$-dimensional standard normal vector, $\boldsymbol{\Sigma}_k^{1/2} = \mathbf{U}_k \boldsymbol{\Lambda}_k^{1/2} \mathbf{U}_k^\top$, and $\mathbf{1}_{\pi_k N}$ is a vector of ones of length $\pi_k N$. The total sample count and mixture proportions were set as

$$N = \begin{cases} 400, & K = 2, \\ 600, & K = 3, \end{cases} \qquad \boldsymbol{\pi} = \begin{cases} [0.5, 0.5], & K = 2, \\ [0.3, 0.33, 0.37], & K = 3. \end{cases}$$

