# OpenReview forum: "Robust Clustering using Gaussian Mixtures in the Presence of Cellwise Outliers"
_TMLR — Accepted by TMLR_

### Review · Reviewer_VrhV · 2025-09-23

**Summary Of Contributions:**

**Summary**

The authors propose a majorization-minimization (MM) algorithm, RobGMM, for learning Gaussian Mixture Models with outliers. The main novelty is that their method accounts for partially corrupted data (cell outliers). The outlier detection is carried out via multiple hypothesis testing (FDR), which in turn provides a penalty term for the likelihood in the MM algorithm.
They test their method in clustering problems with synthetic data with manual corruption and in a real world dataset. Their method consistently outperforms other robust learning methods in several metrics.

**Strengths**

- The paper is clearly written and well structured.

- The motivation (clustering under cell outliers) is relevant and their math is solid.

**Weaknesses**

- RobGMM has a complexity per-iteration of $\mathcal{O}(KNp^4+K^2N)$, where $K$ is the number of mixture components, $N$ the number of data points and $p$ the problem's dimension. A dependence of $p^4$ is considerably high, and probably makes the method unfeasible for high-dimensional data. This doesn't invalidate their method, but at least a discussion of its limitations or a proposal for future improvements would be pertinent.

- As far as I understand, in the experiments they compare RobGMM to robust learning techniques that, in their terminology, consider casewise outliers, while their method detects cellwise outliers. This might be a case of comparing apples and oranges. Is there any other method that considers cellwise outliers? If yes, why not use it as a benchmark? If not, the novelty should be emphasized. I am not familiar with the literature, but I feel the paper lacks more discussion of cellwise outlier detection methods.

**Additional Comments:**

I have a couple of related questions regarding the experimental setup:

- In the synthetic data experiment, the choice of dimension 2 is justified by visualization. However, since your algorithm detects cell-wise outliers, isn't dimension 2 too small for exploiting the ability of RobGMM for detecting outlier dimensions?

- Furthermore, as far as I understand, you create outliers by sampling uniformly distributed data in [-20, 20]. If I understood well, this doesn't create cell-wise outliers, but case-wise outliers. I feel the distinction between both and the ability of your algorithm to detect the former is not well highlighted by this experiment. I acknowledge this is done with the *Top Gear* dataset, but in that case it is not clear how you perturb the "1% randomly selected cells".

- This is just out of curiosity: In the synthetic experiment, how do you visualize the outliers (for example in figure 1f)? I mean, given $(x,y)$, RobGMM might discard $x$ but keep $y$, are the points in 1f only those with no outlier dimension? In general, how often does RobGMM discard a complete data point by discarding each dimension? I guess this is controlled by $\eta_t$, but it seems an interesting metric to assess the efficiency of the algorithm.

**Audience:**

Yes

**Audience Explanation:**

Clustering contaminated data is an important real-world problem, and the paper addresses a difficult version of the problem (partially corrupted data).

**Claims And Evidence:**

Yes

**Claims Explanation:**

The mathematical derivations and experimental setup are rigorous.

**Requested Changes:**

Overall, this is a good paper worth publishing. However, I believe addressing the **Weaknesses** I mentioned above would significantly strengthen the submission.

---

> ### Author Response · Authors · 2025-11-11
> **Response to Reviewer VrhV**
>
> **Comment:**
> *RobGMM has a complexity per-iteration of* $\mathcal{O}\left(KNp^4 + K^2N\right)$, *where* $K$ *is the number of mixture components,* $N$ *the number of data points and* $p$ *the problem's dimension. A dependence of* $p^4$ *is considerably high, and probably makes the method unfeasible for high-dimensional data. This doesn't invalidate their method, but at least a discussion of its limitations or a proposal for future improvements would be pertinent.*
>
> **Response:**
> We thank the reviewer for raising this important point. We agree that the $\mathcal{O}(KNp^4 + K^2N)$ complexity may limit scalability for large $p$. The dominant $\mathcal{O}(p^4)$ term originates from the outlier detection step, specifically from repeatedly computing $(\mathbf{B}_t^{\top}\boldsymbol{\Sigma}_k\mathbf{B}_t)^{-1}$ as $\mathbf{B}_t$ changes.
>
> For a given $\mathbf{B}_t$, the inverse of the covariance submatrix $(\mathbf{B}_t^{\top}\boldsymbol{\Sigma}_k\mathbf{B}_t)$ costs $\mathcal{O}(p^3)$. However, during the outlier detection stage, the covariances $\\{ \boldsymbol{\Sigma}_k \\}$ remain fixed, and $\mathbf{B}_t$ changes only by one column at a time. By leveraging this structure, we can employ *rank-one update and downdate formulas* to avoid recomputing the full inverse. Specifically, using the Sherman–Morrison–Woodbury identity and the Schur complement (block inversion lemma) [1–5], each inverse update can be reduced from $\mathcal{O}(p^3)$ to $\mathcal{O}(p^2)$. This reduces the per-iteration cost by one order in $p$, making the approach more practical for moderately high-dimensional data.
>
> Let
>
> $$
> \mathbf{H} = (\mathbf{B}_t^{\top}\boldsymbol{\Sigma}_k\mathbf{B}_t)^{-1},
> $$
>
> where $\boldsymbol{\Sigma}_k \in \mathbb{R}^{p \times p}$ is symmetric positive definite and $\mathbf{B}_t \in \mathbb{R}^{p \times p_t}$ is a selection matrix composed of $p_t$ columns of the identity matrix $\mathbf{I}_p$. Denoting the selected index set by $\mathcal{S}$, we have
>
> $$
> \mathbf{H} = \left({(\boldsymbol{\Sigma_k})}_{\mathcal{S},\mathcal{S}}\right)^{-1}.
> $$
>
> When one index is added $(\mathcal{S}' = \mathcal{S} \cup \\{i\\})$, the updated inverse is obtained as
>
> $$
> \left({(\boldsymbol{\Sigma_k})}_{\mathcal{S}',\mathcal{S}'}\right)^{-1} =
> \begin{bmatrix}
> \mathbf{H} + \mathbf{H}\boldsymbol{\beta}\alpha^{-1}\boldsymbol{\beta}^{\top}\mathbf{H} & -\mathbf{H}\boldsymbol{\beta}\alpha^{-1} \\\\
> -\alpha^{-1}\boldsymbol{\beta}^{\top}\mathbf{H} & \alpha^{-1}
> \end{bmatrix},
> $$
>
> where $\boldsymbol{\beta} = {(\boldsymbol{\Sigma_k})}_{\mathcal{S},i}$
>
> and $\alpha = {(\boldsymbol{\Sigma_k})}_{i,i} - \boldsymbol{\beta}^{\top}\mathbf{H}\boldsymbol{\beta}$.
>
> Conversely, when one index is removed, we start from the inverse corresponding to the larger set $\mathcal{S}'$,
>
> $$
> \mathbf{H}' =
> \left({(\boldsymbol{\Sigma_k})}_{\mathcal{S}',\mathcal{S}'}\right)^{-1} =
> \begin{bmatrix}
> \mathbf{F} & \boldsymbol{\gamma} \\\\
> \boldsymbol{\gamma}^{\top} & \omega
> \end{bmatrix},
> $$
>
> where $\mathbf{F} \in \mathbb{R}^{p_t \times p_t}$ is the submatrix of $\mathbf{H}'$ corresponding to the retained indices $\mathcal{S}$, $\boldsymbol{\gamma} \in \mathbb{R}^{p_t}$ represents the cross-term between the retained and the removed index, and $\omega \in \mathbb{R}$ is the scalar entry associated with the removed index. Then, the inverse for the reduced index set $\mathcal{S} = \mathcal{S}' \setminus \\{i\\}$ is obtained as
>
> $$
> \left({(\boldsymbol{\Sigma_k})}_{\mathcal{S},\mathcal{S}}\right)^{-1}
> = \mathbf{F} - \frac{1}{\omega}\boldsymbol{\gamma}\boldsymbol{\gamma}^{\top}.
> $$
>
> Both updates involve matrix–vector multiplications and outer products, each costing $\mathcal{O}(p_t^2)$. Since $p_t \le p$, the per-update complexity is upper-bounded by $\mathcal{O}(p^2)$. Consequently, the computational cost of the outlier detection step reduces from $\mathcal{O}(KNp^4)$ to $\mathcal{O}(KNp^3)$, and the total complexity of the proposed method becomes $\mathcal{O}(KNp^3 + K^2N)$. We have added this discussion in the revised manuscript.
>
> ---
>
> **References**
>
> [1] J. Sherman and W. J. Morrison, “Adjustment of an inverse matrix corresponding to a change in one element of a given matrix,” *Ann. Math. Stat.*, 1950.
> [2] M. A. Woodbury, “Inverting modified matrices,” *Statistical Research Group, Princeton University*, 1950.
> [3] P. Stoica and R. L. Moses, *Spectral Analysis of Signals*, Prentice-Hall, 2005.
> [4] F. Zhang (Ed.), *The Schur Complement and Its Applications*, Springer, 2005.
> [5] D. A. Harville, *Matrix Algebra from a Statistician’s Perspective*, Springer, 1997.

---

> ### Author Response · Authors · 2025-11-11
> **Response to Reviewer VrhV**
>
> **Comment:**
> *As far as I understand, in the experiments they compare RobGMM to robust learning techniques that, in their terminology, consider casewise outliers, while their method detects cellwise outliers. This might be a case of comparing apples and oranges. Is there any other method that considers cellwise outliers? If yes, why not use it as a benchmark? If not, the novelty should be emphasized. I am not familiar with the literature, but I feel the paper lacks more discussion of cellwise outlier detection methods.*
>
> **Response:**
> To the best of our knowledge, there are no cellwise outlier detection–based clustering techniques in the literature. Therefore, we chose to compare our methodology with some of the most popular robust clustering techniques such as TCLUST and the Student’s $t$ mixture model. Although TCLUST and the Student’s $t$ mixture model do not explicitly find the cell-outlier positions, they mitigate the effect of cell-outliers and perform clustering to obtain the parameters $\left(\\{ \boldsymbol{\mu}_k , \boldsymbol{\Sigma}_k \\} \right)$. Thus, we believe it is fair to compare the clustering performances of TCLUST and the Student’s $t$ mixture model with the proposed algorithm.

---

> ### Author Response · Authors · 2025-11-11
> **Response to Reviewer VrhV**
>
> **Additional Comment:**
> *In the synthetic data experiment, the choice of dimension 2 is justified by visualization. However, since your algorithm detects cell-wise outliers, isn't dimension 2 too small for exploiting the ability of RobGMM for detecting outlier dimensions?*
>
> **Response:**
> We have now performed the experiments with different choices of problem dimension ($p \in \\{5, 10, 25\\}$). We generated the synthetic data for $K \in \\{2,3\\}$ clusters. Each cluster is modeled as a multivariate Gaussian distribution with distinct mean and covariance structures, constructed as follows.
>
> **Cluster Means:**
> For each feature dimension $d \in \\{1, \dots, p\\}$, the mean values across $K$ clusters were assigned as a random permutation of a fixed set of base mean values, ensuring that the cluster centers remain well separated in every dimension:
>
> $$
> \mathbf{M} =
> \begin{bmatrix}
> \boldsymbol{\mu}_1 & \boldsymbol{\mu}_2 & \cdots & \boldsymbol{\mu}_K
> \end{bmatrix}
> \in \mathbb{R}^{p \times K},
> $$
>
> $$
> \mu_{dk} \in
> \begin{cases}
> \mathrm{Perm}\\{5, 25\\}, & K = 2, \\\\
> \mathrm{Perm}\\{5, 20, 35\\}, & K = 3.
> \end{cases}
> $$
>
>
> **Cluster Covariances:**
> For each cluster $k$, a symmetric positive definite covariance matrix $\boldsymbol{\Sigma}_k \in \mathbb{R}^{p \times p}$ was generated as
>
> $$
> \boldsymbol{\Sigma}_k = \mathbf{U}_k \boldsymbol{\Lambda}_k \mathbf{U}_k^{\top},
> $$
>
> where $\mathbf{U}_k$ denotes the eigenvector matrix obtained from the eigendecomposition of a random symmetric matrix
>
> $$
> \mathbf{R}_k = \mathbf{A}_k \mathbf{A}_k^{\top},
> $$
>
> $$
> [\mathbf{A_k}]_{i,j} \sim \mathcal{N}(0,1).
> $$
>
> The diagonal eigenvalue matrix $\boldsymbol{\Lambda_k} = \mathrm{diag}(\lambda_{k1}, \ldots, \lambda_{kp})$ contains independent entries drawn from uniform distributions with cluster-specific spectral ranges:
>
> $$
> \lambda_{ki} \sim
> \begin{cases}
> \mathcal{U}(6, 15), & k = 1, \\\\
> \mathcal{U}(4, 10), & k = 2, \\\\
> \mathcal{U}(3, 8), & k = 3~(\text{if } K=3).
> \end{cases}
> $$
>
> This procedure yields full-rank, well-conditioned covariance matrices with diverse eigenvalue spectra, thereby producing clusters of distinct orientation and spread.
>
> **Sample Generation:**
> For each cluster $k$, data samples were generated as
>
> $$
> \mathbf{X_k} = \boldsymbol{\mu_k} \mathbf{1}_{\pi_k N}^{\top}+ \boldsymbol{\Sigma}_k^{1/2} \mathbf{Z}_k,\qquad
> \mathbf{Z}_k \in \mathbb{R}^{p \times \pi_k N},
> $$
>
> $$
> \mathbf{z}_{k,t} \sim \mathcal{N}(\mathbf{0}, \mathbf{I}_p),
> $$
>
> where each column $\mathbf{z}_{k,t}$ of $\mathbf{Z}_k$ represents an independent $p$-dimensional standard normal vector, $\boldsymbol{\Sigma_k}^{1/2} = \mathbf{U_k} \boldsymbol{\Lambda_k}^{1/2} \mathbf{U_k}^{\top}$,
>
> and $\mathbf{1}_{\pi_k N}$ is a vector of ones of length $\pi_k N$. The total sample count and mixture proportions were set as
>
> $$
> N =
> \begin{cases}
> 400, & K = 2, \\\\
> 600, & K = 3,
> \end{cases}
> \qquad
> \boldsymbol{\pi} =
> \begin{cases}
> [0.5, 0.5], & K = 2, \\\\
> [0.3, 0.33, 0.37], & K = 3.
> \end{cases}
> $$
>
> The results showing the NRMSE for the estimated means and covariances, as well as the KL divergence with respect to the outlier percentage, are presented in the supplementary material, together with tables summarizing the average computation time (in seconds) of each method across different outlier percentages.
>
> *(See supplementary figures and tables in [large_dimension_simulations.pdf](https://drive.google.com/file/d/11YZXiCQ2L18P5jQzQCsanMHjL0UPuPgW/view?usp=sharing))*

---

> ### Author Response · Authors · 2025-11-11
> **Response to Reviewer VrhV**
>
> **Additional Comment:**
> *Furthermore, as far as I understand, you create outliers by sampling uniformly distributed data in* $[-20, 20]$. *If I understood well, this doesn't create cell-wise outliers, but case-wise outliers. I feel the distinction between both and the ability of your algorithm to detect the former is not well highlighted by this experiment. I acknowledge this is done with the Top Gear dataset, but in that case it is not clear how you perturb the “$1\\%$ randomly selected cells”.*
>
> **Response:**
> The proposed method detects cell-wise outliers: it identifies outliers at the *cell* level when a sample is only partially corrupted and uses the remaining uncorrupted cells for parameter estimation. When all cells of a sample are corrupted, the method discards that entire sample.
>
> For the Top Gear dataset, we inject *cell-wise* contamination by randomly selecting $1\\%$ of the cells (independently for each dimension) and perturbing only those entries, while keeping all other cells of the data sample unchanged. This generation procedure has been clarified in the revised manuscript to emphasize that the synthetic corruption is cell-wise rather than case-wise.

---

> ### Author Response · Authors · 2025-11-11
> **Response to Reviewer VrhV**
>
> **Additional Comment:**
> *This is just out of curiosity: In the synthetic experiment, how do you visualize the outliers (for example in figure 1f)? I mean, given $(x,y)$, RobGMM might discard $x$ but keep $y$, are the points in 1f only those with no outlier dimension? In general, how often does RobGMM discard a complete data point by discarding each dimension? I guess this is controlled by $\eta_t$, but it seems an interesting metric to assess the efficiency of the algorithm.*
>
> **Response:**
> The proposed method discards a sample only when all its cells are corrupted. In the experiment, although the proposed method utilizes the uncorrupted cells of partially corrupted samples for parameter estimation, we excluded both fully and partially corrupted samples from Figure 1(f) for clear visualization.

---

> > ### Comment · Reviewer_VrhV · 2025-11-17
> >
> > I thank the authors for the thorough response. As expected, the experiments with larger dimensions (e.g. p=25) show a huge  computational overhead compared to standard methods. The reduction to cubic dependence is a good starting point but probably not enough for practical applications. Anyway, I also think that the paper makes meaningful progress on the hard problem of learning with cell-wise outliers, which can potentially stimulate further developments. For this reason I am leaning towards accept, provided all the promised changes are incorporated to the final version.

---

### Review · Reviewer_a5w7 · 2025-09-29

**Summary Of Contributions:**

This paper proposes a novel algorithm for Gaussian Mixture Model (GMM)-based clustering that is robust to outliers. This is accomplished by introducing additional variables that index whether a data point is an outlier; specifically, which output dimensions for each data point are deemed to be outliers. The proposed likelihood function is minimized using a majorization minimization framework, with the resulting algorithm, RobGMM, having a computational complexity that is quadratic in the number of clusters ($K)$ and quartic in the output dimension ($p$).

RobGMM is benchmarked against TCLUST and the Student’s t-mixture model on several synthetic datasets, with RobGMM shown to outperform both existing methods. Additionally, experiments on the “Top Gear” dataset demonstrate the capability of RobGMM in detecting outliers while clustering the data into interpretable clusters.

The proposed algorithm addresses a well-known issue of outliers when performing clustering. Outliers often distort the parameters of the GMM, which renders the inferred model to be less useful for clustering on unseen data. The relative simplicity of RobGMM, and optimizing the objective function by means of majorization-minimization, makes the algorithm appealing for users. The demonstrated performance on the chosen datasets also lend some credence to the fact that RobGMM does a better job of inferring the GMM parameters in the presence of cell outliers.

While the relative simplicity of the proposed approach is appreciated, I think the authors have incorrectly justified the choice of thresholds ($_\eta _t$) by framing it as controlling the false discovery rate (FDR) through the Benjamini & Hochberg procedure. In a multiple-hypothesis testing framework, the hypotheses have to be set beforehand; here, the choice of thresholds is in effect, setting the hypotheses based on the significance levels – this is likely p-hacking. Hence, an alternate justification has to be provided for this design choice.
In addition to this, the experiments are a little lacking in several key aspects:

1.	How were the specific Gaussian parameters chosen for the synthetic experiments?
2.	Additional experiments on higher-dimension datasets are needed for an extensive validation of RobGMM. The various algorithms should also be compared by time complexity as RobGMM scales quartically with output dimension.
3.	The chosen numerical metrics appear to account for the accuracy of outlier detection as another cluster.

   a.	Does the Student’s t-mixture model also include an additional outlier cluster? If not, this unfairly penalizes that model.

   b.	The accuracy of outlier detection should probably be separated from that of the clusters. Poor outlier detection with accurate GMM parameter estimation could be an acceptable trade-off, especially when using heavy-tailed distribution models.
4.	Why was only RobGMM was run for the real dataset?

**Audience:**

Yes

**Audience Explanation:**

The paper approaches a known problem of clustering -- the presence of outliers. A novel approach to GMM-based clustering that is robust to outliers while also detecting outliers is a valuable contribution.

**Broader Impact Concerns:**

I do not have any broader impact concerns with this work.

**Claims And Evidence:**

Yes

**Claims Explanation:**

The experimental evidence does support the claims made but the justification for the thresholds has to be corrected.

**Requested Changes:**

**Critical changes:**

1.	Pursuant to the comment in the previous section, an alternative justification has to be made for how the thresholds in the hypotheses were selected.
2.	More extensive experimental validation of RobGMM is required to more fully demonstrate that the proposed approach is utile for robust GMM clustering across a variety of settings, and comparable, if not better, than existing methods.
3.	Clustering of the Top Gear dataset using RobGMM should be compared to that by other robust clustering approaches.

**Suggested changes:**

1.	The notation in (11) is confusing. The summation over $t$ runs to $N_i$ which itself is a sum over $N$ terms. This may be incorrect too, since $eta_N$ may never be added as a penalty but $b_i(N)$ contributes to some $N_i$.
2.	Typo: First line of Section 4 should read “For the reader’s convenience”.
3.	More methods should be used as benchmarks for a comprehensive evaluation of RobGMM’s performance.
4.	It is not clear in Figure 1 whether the various algorithms detect outliers. Also, the clusters for all the algorithms seem to be very similar.
5.	The first scenario described in Section 5.2 lacks details.

  a.	The avoidance of why outlier detections should not occur in the 99-th percentile of any cluster should be better explained.

  b.	How were the values shifted?

---

> ### Author Response · Authors · 2025-11-11
> **Response to Reviewer a5w7**
>
> **Comment:**
> *How were the specific Gaussian parameters chosen for the synthetic experiments?*
>
> **Response:**
> Synthetic datasets of dimension $p=2$ (now extended to higher dimensions $p \in \\{5, 10, 25\\}$) were generated for $K \in \\{2,3\\}$ Gaussian clusters, each characterized by distinct mean and covariance structures.
> For each feature dimension $d$, the cluster means were assigned as random permutations of fixed base mean values, for example, $\\{5, 25\\}$ for $K=2$ and $\\{5, 20, 35\\}$ for $K=3$, forming the mean matrix $\mathbf{M} = [\boldsymbol{\mu}_1, \ldots, \boldsymbol{\mu}_K] \in \mathbb{R}^{p \times K}$, where each $\boldsymbol{\mu}_k$ denotes the $p$-dimensional mean vector of cluster $k$. In the experiment with four clusters, we used a set of base mean values comprising four distinct values.
>
> The covariance matrix of each cluster was constructed as
>
> $$
> \boldsymbol{\Sigma}_k = \mathbf{U}_k \boldsymbol{\Lambda}_k \mathbf{U}_k^{\top},
> $$
>
> where $\mathbf{U}_k$ contains the eigenvectors of a random symmetric matrix
>
> $$
> \mathbf{R}_k = \mathbf{A}_k \mathbf{A}_k^{\top},
> $$
>
> $$
> [\mathbf{A_k}]_{i,j} \sim \mathcal{N}(0,1).
> $$
>
> The eigenvalues $\{\lambda_{ki}\}$ were sampled uniformly from cluster-specific spectral intervals, namely $\mathcal{U}(6,15)$, $\mathcal{U}(4,10)$, and $\mathcal{U}(3,8)$ for $k=1,2,3$, respectively, ensuring full-rank, well-conditioned covariance structures with varying anisotropy. For the experiment with four clusters, the covariance matrices were generated following the same procedure.
>
> Samples were generated as
>
> $$
> \mathbf{X_k} = \boldsymbol{\mu_k} \mathbf{1}_{\pi_k N}^{\top}+ \boldsymbol{\Sigma}_k^{1/2} \mathbf{Z}_k,
> $$
>
> $$
> \mathbf{Z_k} \in \mathbb{R}^{p \times \pi_k N}, \quad
> \mathbf{z}_{k,t} \sim \mathcal{N}(\mathbf{0}, \mathbf{I}_p),
> $$
>
> where each column $\mathbf{z}_{k,t}$ of $\mathbf{Z_k}$ represents an independent $p$-dimensional standard normal vector.
> Here, $\boldsymbol{\Sigma_k}^{1/2} = \mathbf{U_k} \boldsymbol{\Lambda_k}^{1/2} \mathbf{U_k}^{\top}$,
>
> and $\mathbf{1}_{\pi_k N}$ is a vector of ones of length $\pi_k N$.
>
> The total sample size and mixture proportions were set to $N = 400$, $\boldsymbol{\pi} = [0.5, 0.5]$ for $K = 2$, $N = 600$, $\boldsymbol{\pi} = [0.3, 0.33, 0.37]$ for $K = 3$, and $N = 400$, $\boldsymbol{\pi} = [0.25, 0.25, 0.25, 0.25]$ for $K = 4$.

---

> ### Author Response · Authors · 2025-11-11
> **Response to Reviewer a5w7**
>
> **Comment:**
> *Additional experiments on higher-dimension datasets are needed for an extensive validation of RobGMM. The various algorithms should also be compared by time complexity as RobGMM scales quartically with output dimension.*
>
> **Response:**
> We agree that the $\mathcal{O}(KNp^4 + K^2N)$ complexity may limit scalability for large $p$. The dominant $\mathcal{O}(p^4)$ term originates from the outlier detection step, specifically from repeatedly computing $(\mathbf{B}_t^{\top}\boldsymbol{\Sigma}_k\mathbf{B}_t)^{-1}$ as $\mathbf{B}_t$ changes.
>
> For a given $\mathbf{B}_t$, the inverse of the covariance submatrix $(\mathbf{B}_t^{\top}\boldsymbol{\Sigma}_k\mathbf{B}_t)$ costs $\mathcal{O}(p^3)$. However, during the outlier detection stage, the covariances $\\{ \boldsymbol{\Sigma}_k \\}$ remain fixed, and $\mathbf{B}_t$ changes only by one column at a time. By leveraging this structure, we can employ *rank-one update and downdate formulas* to avoid recomputing the full inverse. Specifically, using the Sherman–Morrison–Woodbury identity and the Schur complement (block inversion lemma) [1–5], each inverse update can be reduced from $\mathcal{O}(p^3)$ to $\mathcal{O}(p^2)$. This reduces the per-iteration cost by one order in $p$, making the approach more practical for moderately high-dimensional data.
>
> Let
>
> $$
> \mathbf{H} = (\mathbf{B}_t^{\top}\boldsymbol{\Sigma}_k\mathbf{B}_t)^{-1},
> $$
>
> where $\boldsymbol{\Sigma}_k \in \mathbb{R}^{p \times p}$ is symmetric positive definite and $\mathbf{B}_t \in \mathbb{R}^{p \times p_t}$ is a selection matrix composed of $p_t$ columns of the identity matrix $\mathbf{I}_p$. Denoting the selected index set by $\mathcal{S}$, we have
>
> $$
> \mathbf{H} = \left({(\boldsymbol{\Sigma_k})}_{\mathcal{S},\mathcal{S}}\right)^{-1}.
> $$
>
> When one index is added $(\mathcal{S}' = \mathcal{S} \cup \\{i\\})$, the updated inverse is obtained as
>
> $$
> \left({(\boldsymbol{\Sigma_k})}_{\mathcal{S}',\mathcal{S}'}\right)^{-1} =
> \begin{bmatrix}
> \mathbf{H} + \mathbf{H}\boldsymbol{\beta}\alpha^{-1}\boldsymbol{\beta}^{\top}\mathbf{H} & -\mathbf{H}\boldsymbol{\beta}\alpha^{-1} \\\\
> -\alpha^{-1}\boldsymbol{\beta}^{\top}\mathbf{H} & \alpha^{-1}
> \end{bmatrix},
> $$
>
> where $\boldsymbol{\beta} = {(\boldsymbol{\Sigma_k})}_{\mathcal{S},i}$
>
> and $\alpha = {(\boldsymbol{\Sigma_k})}_{i,i} - \boldsymbol{\beta}^{\top}\mathbf{H}\boldsymbol{\beta}$.
>
> Conversely, when one index is removed, we start from the inverse corresponding to the larger set $\mathcal{S}'$,
>
> $$
> \mathbf{H}' =
> \left({(\boldsymbol{\Sigma_k})}_{\mathcal{S}',\mathcal{S}'}\right)^{-1} =
> \begin{bmatrix}
> \mathbf{F} & \boldsymbol{\gamma} \\\\
> \boldsymbol{\gamma}^{\top} & \omega
> \end{bmatrix},
> $$
>
> where $\mathbf{F} \in \mathbb{R}^{p_t \times p_t}$ is the submatrix of $\mathbf{H}'$ corresponding to the retained indices $\mathcal{S}$, $\boldsymbol{\gamma} \in \mathbb{R}^{p_t}$ represents the cross-term between the retained and the removed index, and $\omega \in \mathbb{R}$ is the scalar entry associated with the removed index. Then, the inverse for the reduced index set $\mathcal{S} = \mathcal{S}' \setminus \\{i\\}$ is obtained as
>
> $$
> \left({(\boldsymbol{\Sigma_k})}_{\mathcal{S},\mathcal{S}}\right)^{-1}
> = \mathbf{F} - \frac{1}{\omega}\boldsymbol{\gamma}\boldsymbol{\gamma}^{\top}.
> $$
>
> Both updates involve matrix–vector multiplications and outer products, each costing $\mathcal{O}(p_t^2)$. Since $p_t \le p$, the per-update complexity is upper-bounded by $\mathcal{O}(p^2)$. Consequently, the computational cost of the outlier detection step reduces from $\mathcal{O}(KNp^4)$ to $\mathcal{O}(KNp^3)$, and the total complexity of the proposed method becomes $\mathcal{O}(KNp^3 + K^2N)$. We have added this discussion in the revised manuscript. We have now performed the experiments for different choices of the number of dimensions ($p=5,10,25$).
>
> ---
>
> **References**
>
> [1] J. Sherman and W. J. Morrison, “Adjustment of an inverse matrix corresponding to a change in one element of a given matrix,” *Ann. Math. Stat.*, 1950.
> [2] M. A. Woodbury, “Inverting modified matrices,” *Statistical Research Group, Princeton University*, 1950.
> [3] P. Stoica and R. L. Moses, *Spectral Analysis of Signals*, Prentice-Hall, 2005.
> [4] F. Zhang (Ed.), *The Schur Complement and Its Applications*, Springer, 2005.
> [5] D. A. Harville, *Matrix Algebra from a Statistician’s Perspective*, Springer, 1997.

---

> ### Author Response · Authors · 2025-11-11
> **Response to Reviewer a5w7**
>
> **Comment:**
> *Does the Student’s t-mixture model also include an additional outlier cluster? If not, this unfairly penalizes that model.*
>
> **Response:**
> The Student’s $t$ mixture model does not include an additional outlier cluster. This is because the outliers in our setting are distributed arbitrarily around and between the data clusters, making it infeasible to represent them as a separate component. Nevertheless, due to the heavy-tailed nature of the $t$ distribution, the Student’s $t$ mixture model inherently downweights cell-wise outliers while estimating the distribution parameters.

---

> ### Author Response · Authors · 2025-11-11
> **Response to Reviewer a5w7**
>
> **Comment:**
> *The accuracy of outlier detection should probably be separated from that of the clusters. Poor outlier detection with accurate GMM parameter estimation could be an acceptable trade-off, especially when using heavy-tailed distribution models.*
>
> **Response:**
> First regarding your comment on *“Poor outlier detection with accurate GMM parameter estimation could be an acceptable trade-off, especially when using heavy-tailed distribution models,”* this is not possible, if the method poorly detects the outliers then the clustering performance degrades significantly, our simulations clearly demonstrate this aspect. Since the clustering is performed on the contaminated data, the performance metrics evaluation would not be practical without the outliers. We now consider two cases: in case 1, the performance metrics are computed by considering 4 clusters and one outlier group, please note that in this case the competing algorithms do not detect outliers. The results obtained for 10% and 20% contamination are shown in Table 1 and Table 2 respectively. In case 2, we compute the performance metrics in which we also empower the competing algorithms to detect outliers as follows. We first implement the competing methods on the corrupted data to estimate the distribution parameters (means, covariances, and mixture proportions) and with these estimates we then run the FDR based multiple hypothesis testing to detect the outliers for the corresponding methods. The obtained results are shown in Table 3 and Table 4. From the results, we see that addition of the outlier cluster to the other robust methods degrades the performance metrics as the poor parameter estimation yields inaccurate outlier detections.
>
> ---
>
> **Table 1.** Performance metrics for RobGMM and other methods for 10% outliers
>
> | Metric | **GMM** (mean ± std. dev.) | **Student's t mixture** (mean ± std. dev.) | **TCLUST** (mean ± std. dev.) | **RobGMM** (mean ± std. dev.) |
> |:-------|:---------------------------:|:------------------------------------------:|:------------------------------:|:------------------------------:|
> | **Accuracy** | 0.708 ± 0.100 | 0.821 ± 0.106 | 0.900 ± 0.048 | 0.981 ± 0.008 |
> | **EMPC**     | 0.378 ± 0.195 | 0.579 ± 0.204 | 0.714 ± 0.099 | 0.949 ± 0.022 |
>
> ---
>
> **Table 2.** Performance metrics for RobGMM and other methods for 20% outliers
>
> | Metric | **GMM** (mean ± std. dev.) | **Student's t mixture** (mean ± std. dev.) | **TCLUST** (mean ± std. dev.) | **RobGMM** (mean ± std. dev.) |
> |:-------|:---------------------------:|:------------------------------------------:|:------------------------------:|:------------------------------:|
> | **Accuracy** | 0.580 ± 0.094 | 0.682 ± 0.079 | 0.718 ± 0.100 | 0.962 ± 0.008 |
> | **EMPC**     | 0.217 ± 0.215 | 0.426 ± 0.169 | 0.467 ± 0.211 | 0.947 ± 0.015 |
>
> ---
>
> **Table 3.** Performance metrics for RobGMM and other methods for 10% outliers
>
> | Metric | **GMM** (mean ± std. dev.) | **Student's t mixture** (mean ± std. dev.) | **TCLUST** (mean ± std. dev.) | **RobGMM** (mean ± std. dev.) |
> |:-------|:---------------------------:|:------------------------------------------:|:------------------------------:|:------------------------------:|
> | **Accuracy** | 0.694 ± 0.110 | 0.809 ± 0.105 | 0.819 ± 0.106 | 0.980 ± 0.006 |
> | **EMPC**     | 0.134 ± 0.220 | 0.344 ± 0.211 | 0.349 ± 0.215 | 0.950 ± 0.017 |
>
> ---
>
> **Table 4.** Performance metrics for RobGMM and other methods for 20% outliers
>
> | Metric | **GMM** (mean ± std. dev.) | **Student's t mixture** (mean ± std. dev.) | **TCLUST** (mean ± std. dev.) | **RobGMM** (mean ± std. dev.) |
> |:-------|:---------------------------:|:------------------------------------------:|:------------------------------:|:------------------------------:|
> | **Accuracy** | 0.542 ± 0.095 | 0.681 ± 0.093 | 0.690 ± 0.092 | 0.972 ± 0.012 |
> | **EMPC**     | -0.063 ± 0.218 | 0.226 ± 0.200 | 0.192 ± 0.198 | 0.948 ± 0.022 |

---

> ### Author Response · Authors · 2025-11-11
> **Response to Reviewer a5w7**
>
> **Comment:**
> *Why was only RobGMM was run for the real dataset?*
>
> **Response:**
> We have now evaluated all the competing methods on the Top Gear dataset in the revised manuscript.

---

> ### Author Response · Authors · 2025-11-11
> **Response to Reviewer a5w7**
>
> **Comment (Critical Changes):**
> *Pursuant to the comment in the previous section, an alternative justification has to be made for how the thresholds in the hypotheses were selected.*
>
> **Response:**
> We thank the reviewer for raising this important point regarding the justification of the threshold values $\eta_t$. We would like to clarify that, in our method, the thresholds $\{\eta_t\}$ are **not chosen adaptively from the data** in the sense implied by p-hacking. Specifically, the $\eta_t$ values are derived directly from the Benjamini–Hochberg (BH) procedure and the chi-square null distribution (Eq. (29)) as fixed quantiles corresponding to the ordered significance levels $\gamma_t = \alpha t / N$. Once $\alpha$ and $N$ are specified, these thresholds are **deterministic and do not depend on the observed data or the estimated model parameters**.
>
> What does change across iterations are the test statistics $T_i(t)$, which depend on the current parameter estimates. At each iteration, we recompute the test statistics and apply the same BH thresholds to decide which cells are flagged as outliers. Thus, while the testing procedure is iterative, the thresholds themselves are not adaptively selected from the data, and there is no selective redefinition of hypotheses based on observed significance — the key feature of p-hacking.

---

> ### Author Response · Authors · 2025-11-11
> **Response to Reviewer a5w7**
>
> **Comment (Critical Changes):**
> *More extensive experimental validation of RobGMM is required to more fully demonstrate that the proposed approach is utile for robust GMM clustering across a variety of settings, and comparable, if not better, than existing methods.*
>
> **Response:**
> We have now performed the experiments with different choices of problem dimension ($p \in \\{5, 10, 25\\}$). We generated the synthetic data for $K \in \\{2,3\\}$ clusters. Each cluster is modeled as a multivariate Gaussian distribution with distinct mean and covariance structures, constructed as follows.
>
> **Cluster Means:**
> For each feature dimension $d \in \\{1, \dots, p\\}$, the mean values across $K$ clusters were assigned as a random permutation of a fixed set of base mean values, ensuring that the cluster centers remain well separated in every dimension:
>
> $$
> \mathbf{M} =
> \begin{bmatrix}
> \boldsymbol{\mu}_1 & \boldsymbol{\mu}_2 & \cdots & \boldsymbol{\mu}_K
> \end{bmatrix}
> \in \mathbb{R}^{p \times K},
> $$
>
> $$
> \mu_{dk} \in
> \begin{cases}
> \mathrm{Perm}\\{5, 25\\}, & K = 2, \\\\
> \mathrm{Perm}\\{5, 20, 35\\}, & K = 3.
> \end{cases}
> $$
>
>
> **Cluster Covariances:**
> For each cluster $k$, a symmetric positive definite covariance matrix $\boldsymbol{\Sigma}_k \in \mathbb{R}^{p \times p}$ was generated as
>
> $$
> \boldsymbol{\Sigma}_k = \mathbf{U}_k \boldsymbol{\Lambda}_k \mathbf{U}_k^{\top},
> $$
>
> where $\mathbf{U}_k$ denotes the eigenvector matrix obtained from the eigendecomposition of a random symmetric matrix
>
> $$
> \mathbf{R}_k = \mathbf{A}_k \mathbf{A}_k^{\top},
> $$
>
> $$
> [\mathbf{A_k}]_{i,j} \sim \mathcal{N}(0,1).
> $$
>
> The diagonal eigenvalue matrix $\boldsymbol{\Lambda_k} = \mathrm{diag}(\lambda_{k1}, \ldots, \lambda_{kp})$ contains independent entries drawn from uniform distributions with cluster-specific spectral ranges:
>
> $$
> \lambda_{ki} \sim
> \begin{cases}
> \mathcal{U}(6, 15), & k = 1, \\\\
> \mathcal{U}(4, 10), & k = 2, \\\\
> \mathcal{U}(3, 8), & k = 3~(\text{if } K=3).
> \end{cases}
> $$
>
> This procedure yields full-rank, well-conditioned covariance matrices with diverse eigenvalue spectra, thereby producing clusters of distinct orientation and spread.
>
> **Sample Generation:**
> For each cluster $k$, data samples were generated as
>
> $$
> \mathbf{X_k} = \boldsymbol{\mu_k} \mathbf{1}_{\pi_k N}^{\top}+ \boldsymbol{\Sigma}_k^{1/2} \mathbf{Z}_k,\qquad
> \mathbf{Z}_k \in \mathbb{R}^{p \times \pi_k N},
> $$
>
> $$
> \mathbf{z}_{k,t} \sim \mathcal{N}(\mathbf{0}, \mathbf{I}_p),
> $$
>
> where each column $\mathbf{z}_{k,t}$ of $\mathbf{Z}_k$ represents an independent $p$-dimensional standard normal vector, $\boldsymbol{\Sigma_k}^{1/2} = \mathbf{U_k} \boldsymbol{\Lambda_k}^{1/2} \mathbf{U_k}^{\top}$,
>
> and $\mathbf{1}_{\pi_k N}$ is a vector of ones of length $\pi_k N$. The total sample count and mixture proportions were set as
>
> $$
> N =
> \begin{cases}
> 400, & K = 2, \\\\
> 600, & K = 3,
> \end{cases}
> \qquad
> \boldsymbol{\pi} =
> \begin{cases}
> [0.5, 0.5], & K = 2, \\\\
> [0.3, 0.33, 0.37], & K = 3.
> \end{cases}
> $$
>
> The results showing the NRMSE for the estimated means and covariances, as well as the KL divergence with respect to the outlier percentage, are presented in the supplementary material, together with tables summarizing the average computation time (in seconds) of each method across different outlier percentages.
>
> *(See supplementary figures and tables in [large_dimension_simulations.pdf](https://drive.google.com/file/d/11YZXiCQ2L18P5jQzQCsanMHjL0UPuPgW/view?usp=sharing))*

---

> ### Author Response · Authors · 2025-11-11
> **Response to Reviewer a5w7**
>
> **Comment (Critical Changes):**
> *Clustering of the Top Gear dataset using RobGMM should be compared to that by other robust clustering approaches.*
>
> **Response:**
> All competing methods have been implemented for clustering on the Top Gear dataset in the revised manuscript. We first contaminate the dataset by introducing $1\\%$ cell outliers independently in each dimension and then apply all the methods to the contaminated data. Subsequently, we detect the cell-outlier positions using the FDR-based multiple hypothesis testing procedure applied to the distribution parameters (means, covariances, and mixture proportions) estimated by each method. The true and detected cell-outlier positions are shown in supplementary material, which demonstrate that the proposed method detects the cell-outlier positions more accurately than the competing methods.
>
> *(See supplementary figures in [all_methods_topGear.pdf](https://drive.google.com/file/d/1QQYhYsHmvRrCxeXhKFZpIl5DpD80p7mR/view?usp=sharing))*

---

> ### Author Response · Authors · 2025-11-11
> **Response to Reviewer a5w7**
>
> **Comment (Suggested Changes):**
> *The notation in (11) is confusing. The summation over $t$ runs to $N_i$ which itself is a sum over $N$ terms. This may be incorrect too, since $\eta_N$ may never be added as a penalty but $b_i(N)$ contributes to some $N_i$.*
>
> **Response:**
> We hope to clarify the notation in (11) via the following explanation. We perform outlier detection via a cyclic approach (over $i$). For the $i^{\text{th}}$ variable, we solve the following problem:
>
> $$
> \underset{N_i, \\{ b_i(t) \\}_{t=1}^{N}}{\min} \mathcal{F},
> $$
>
> where
>
> $$
> \mathcal{F} = \sum_{t=1}^{N} b_i(t) T_i(t) + \sum_{t=1}^{N_i} \eta_t,
> $$
>
>
>
>
> to determine $N_i$ and $\{b_i(t)\}$. We test the criterion for all possible values of $N_i$ (i.e., $N_i \in [0, N]$). Hence, $\eta_N$ will indeed appear in the penalty when solving the above problem. We hope this clarifies Equation (11).

---

> ### Author Response · Authors · 2025-11-11
> **Response to Reviewer a5w7**
>
> **Comment (Suggested Changes):**
> *Typo: First line of Section 4 should read “For the reader’s convenience”.*
>
> **Response:**
> Corrected.

---

> ### Author Response · Authors · 2025-11-11
> **Response to Reviewer a5w7**
>
> **Comment (Suggested Changes):**
> *More methods should be used as benchmarks for a comprehensive evaluation of RobGMM’s performance.*
>
> **Response:**
> The proposed RobGMM was compared with well-established robust clustering methods, namely TCLUST and the Student’s $t$ mixture model, along with the standard Gaussian mixture model. The competing methods do not explicitly detect cell-wise outliers; however, TCLUST and the Student’s $t$ mixture model implicitly account for their effect during parameter estimation. To the best of our knowledge, there are no robust clustering methods in the literature that consider cell outliers.

---

> ### Author Response · Authors · 2025-11-11
> **Response to Reviewer a5w7**
>
> **Comment (Suggested Changes):**
> *It is not clear in Figure 1 whether the various algorithms detect outliers. Also, the clusters for all the algorithms seem to be very similar.*
>
> **Response:**
> The competing methods do not explicitly detect cell-wise outliers; however, TCLUST and the Student’s $t$ mixture model implicitly account for their effect during parameter estimation.

---

> ### Author Response · Authors · 2025-11-11
> **Response to Reviewer a5w7**
>
> **Comment (Suggested Changes):**
> *The avoidance of why outlier detections should not occur in the 99-th percentile of any cluster should be better explained.*
>
> **Response:**
> We have clarified in the revised manuscript that the restriction beyond the 99th percentile applies only to the synthetic outlier generation procedure, which was designed to ensure that the generated outliers lie outside the genuine cluster regions. The outlier detection in RobGMM is not constrained by this condition and operates independently based on the estimated model parameters.

---

> ### Author Response · Authors · 2025-11-11
> **Response to Reviewer a5w7**
>
> **Comment (Suggested Changes):**
> *How were the values shifted?*
>
> **Response:**
> Outliers were generated by first randomly selecting cell indices from the data matrix and then shifting their values by random perturbations that are uniformly distributed. In some cases, small perturbations could keep the outlier candidates within the cluster boundaries. Therefore, after each shift, we verify whether the distance of the perturbed value from all cluster means exceeds three standard deviations of the respective clusters. If not, the perturbation value is re-sampled until all outliers are shifted outside the cluster regions, ensuring that they lie beyond the 99th percentile of any cluster. The outlier detection in RobGMM, however, is not constrained by this condition and operates independently based on the estimated model parameters.

---

### Review · Reviewer_LrAE · 2025-10-27

**Summary Of Contributions:**

The paper introduces an algorithm for performing clustering and parameter estimation over K-dimensional multivariate Gaussian mixture models. The key challenge faced in the manuscript is the presence of what is called *cellwise outliers* (deliberate or non-deliberate corrupted features of observations in the dataset), which the authors consider to be around 1%-10% of their experiments. The modeling choice in the paper is to introduce a sort of hot-encoding binary variables, which mask or unmask the presence of outliers per data-point. In practice, this allows the selection of the sub-covariance matrices and sub-vectors of the Gaussian multivariate parameters to be estimated precisely. The learning algorithm is based on what is described as majorization-maximization (MM), which the current reviewer is not familiar with. In the end, after every step where hypothesis-testing (via false discovery rate of FDR) is made to update the beliefs on the presence of outliers via the binary variables, a constrained-type of optimization is done in closed form for the GMM parameters (mixing weights, means, covariances). Experiments are first conducted for a synthetic two-dim dataset, second for another with different outlier % rates, and third for another real-world dataset.

**Additional Comments:**

### Minor issues
1. I would've moved the numbers and settings of the numerical studies in pp. 10 and pp.13 to the appendix, which is the standard practice for venues like TMLR.

**Audience:**

No

**Audience Explanation:**

However, I still have some concerns and *detected* points of weaknesses that I would like to highlight here:

1) GMMs have been for at least *two* decades a key milestone and *starting point* for so many probabilistic modelling problems and challenges. Since the 90s, the EM algorithm stands out as one of the standard solutions, at its easy adaptation to non-Gaussian likelihoods, missing data, and optimization-based learning of parameters have defined a strong SOTA baseline that one should not ignore. In this regard, I see the current proposed work limited due to the lack of comparison and connection with such line of works. The problem introduced is somehow *niched*  to some potential application that is not entirely clear for the reader. Later, one quickly sees the problems that the model could have in three directions: i) larger dimensionality $p$, ii) larger rate of outliers and iii) consideration of non-Gaussian distributed data.
2) For my taste as reviewer, the paper is somehow limited on the empirical results, and the exploration of the method performance. The synthetic data presentation is a bit *naive* with the original parameters and hyperparameters shown in pp. 13 for later discovering them. Whenever the method is relatively simple or easy to compute/analyse, one in the ML/AI community should expect larger experiments and results so the performance is characterised at a max level.
3) The experiments and their results shown in Fig. 2 are OK to me, but inevitably remind me to the ones from Figure 1 in Ghahramani & Jordan (1993) where the solution is tested to rates ranging from 1% up to 100%. The way the paper addressed the problem is in many ways similar to such classical method, with EM used to estimate/fill missing data in a robust manner. The lack of discussion and the similarity of the proposed algorithm (taking the definition written in the beginning of section 4.1 pp.5) concern me significantly.
4) The computational cost is correctly characterised, but I am highly concerned about its feasibility in real-world studies or methods with such super-cubical complexity on the vector parameters.
5) Last but no least, I miss some references to well-known ML/AI outlier detection methods, and latent-variables (that's what $b$ is somehow, but without prior or admitted randomness). The way it is modelled with such masks, remind me to this work from D. Blei's lab some years ago: https://arxiv.org/pdf/1606.03860. I am aware that the scope is quite different, but still useful for the faced problems in the manuscript.

---
#### References:
- _Zoubin Ghahramani, Michael I. Jordan_ (NIPS 1993) *Supervised learning from incomplete data via an EM approach.*

**Claims And Evidence:**

Yes

**Claims Explanation:**

To me, the main strengths of the paper are:

1) Closed-form solution for the estimation of the GMM parameters, and the exact computation of the computational cost for the algorithm --- which, unfortunately, is $p^{4}$ for $p$ being the dimension of the Gaussian components.
2) The choice of using an additional *masking* variable to indicate/estimate the presence of outliers. I do believe it one of the correct ways to model this problem, and it has been longly considered in the literature.
3) The introduction of the hypothesis-testing (FDR) method is interesting, and I liked how it combines with the maximization algorithm. However, I was not particularly familiar with this sort of solution for the GMM learning problem. As reviewer, I am usually more experienced in the EM-based solutions which have dominated the GMM problem since the 90s.
4) Despite it is limited to a relatively small experiment with synthetic data, the performance included in Table 1 is impressive and a good sign that some modelling choices are correctly taken. I would've like to see the method tested in cases with misspecified $K$ for instance.

**Requested Changes:**

I would like to see additional progress on the main points I raised as weaknesses in my review. From all of theses ones, I would prioritize the EM connection and revision, which is perhaps the usual point-of-reference for readers familiar with GMMs. However, I also want to be honest in the sense that I see the current manuscript significantly under the TMLR threshold, parrticularly on the interest/fitting side with the potential readers/researchers/practitioners which focus on TMLR published works. In the end, the paper finds a solution, which does not completely scales or adapts to non-Gaussian problems for a GMM problem, well-known in the community and analyzed since the 90s in multiple ways not exactly revised in the paper.

---

> ### Author Response · Authors · 2025-11-11
> **Response to Reviewer LrAE**
>
> **Comment:**
> *GMMs have been for at least two decades a key milestone and starting point for so many probabilistic modelling problems and challenges. Since the 90s, the EM algorithm stands out as one of the standard solutions, at its easy adaptation to non-Gaussian likelihoods, missing data, and optimization-based learning of parameters have defined a strong SOTA baseline that one should not ignore. In this regard, I see the current proposed work limited due to the lack of comparison and connection with such line of works. The problem introduced is somehow niched to some potential application that is not entirely clear for the reader. Later, one quickly sees the problems that the model could have in three directions: (i) larger dimensionality, (ii) larger rate of outliers, and (iii) consideration of non-Gaussian distributed data.*
>
> **Response (Point (i)):**
>
> We agree that the $\mathcal{O}(KNp^4 + K^2N)$ complexity may limit scalability for large $p$. The dominant $\mathcal{O}(p^4)$ term originates from the outlier detection step, specifically from repeatedly computing $(\mathbf{B}_t^{\top}\boldsymbol{\Sigma}_k\mathbf{B}_t)^{-1}$ as $\mathbf{B}_t$ changes.
>
> For a given $\mathbf{B}_t$, the inverse of the covariance submatrix $(\mathbf{B}_t^{\top}\boldsymbol{\Sigma}_k\mathbf{B}_t)$ costs $\mathcal{O}(p^3)$. However, during the outlier detection stage, the covariances $\\{ \boldsymbol{\Sigma}_k \\}$ remain fixed, and $\mathbf{B}_t$ changes only by one column at a time. By leveraging this structure, we can employ *rank-one update and downdate formulas* to avoid recomputing the full inverse. Specifically, using the Sherman–Morrison–Woodbury identity and the Schur complement (block inversion lemma) [1–5], each inverse update can be reduced from $\mathcal{O}(p^3)$ to $\mathcal{O}(p^2)$. This reduces the per-iteration cost by one order in $p$, making the approach more practical for moderately high-dimensional data.
>
> Let
>
> $$
> \mathbf{H} = (\mathbf{B}_t^{\top}\boldsymbol{\Sigma}_k\mathbf{B}_t)^{-1},
> $$
>
> where $\boldsymbol{\Sigma}_k \in \mathbb{R}^{p \times p}$ is symmetric positive definite and $\mathbf{B}_t \in \mathbb{R}^{p \times p_t}$ is a selection matrix composed of $p_t$ columns of the identity matrix $\mathbf{I}_p$. Denoting the selected index set by $\mathcal{S}$, we have
>
> $$
> \mathbf{H} = \left({(\boldsymbol{\Sigma_k})}_{\mathcal{S},\mathcal{S}}\right)^{-1}.
> $$
>
> When one index is added $(\mathcal{S}' = \mathcal{S} \cup \\{i\\})$, the updated inverse is obtained as
>
> $$
> \left({(\boldsymbol{\Sigma_k})}_{\mathcal{S}',\mathcal{S}'}\right)^{-1} =
> \begin{bmatrix}
> \mathbf{H} + \mathbf{H}\boldsymbol{\beta}\alpha^{-1}\boldsymbol{\beta}^{\top}\mathbf{H} & -\mathbf{H}\boldsymbol{\beta}\alpha^{-1} \\\\
> -\alpha^{-1}\boldsymbol{\beta}^{\top}\mathbf{H} & \alpha^{-1}
> \end{bmatrix},
> $$
>
> where $\boldsymbol{\beta} = {(\boldsymbol{\Sigma_k})}_{\mathcal{S},i}$
>
> and $\alpha = {(\boldsymbol{\Sigma_k})}_{i,i} - \boldsymbol{\beta}^{\top}\mathbf{H}\boldsymbol{\beta}$.
>
> Conversely, when one index is removed, we start from the inverse corresponding to the larger set $\mathcal{S}'$,
>
> $$
> \mathbf{H}' =
> \left({(\boldsymbol{\Sigma_k})}_{\mathcal{S}',\mathcal{S}'}\right)^{-1} =
> \begin{bmatrix}
> \mathbf{F} & \boldsymbol{\gamma} \\\\
> \boldsymbol{\gamma}^{\top} & \omega
> \end{bmatrix},
> $$
>
> where $\mathbf{F} \in \mathbb{R}^{p_t \times p_t}$ is the submatrix of $\mathbf{H}'$ corresponding to the retained indices $\mathcal{S}$, $\boldsymbol{\gamma} \in \mathbb{R}^{p_t}$ represents the cross-term between the retained and the removed index, and $\omega \in \mathbb{R}$ is the scalar entry associated with the removed index. Then, the inverse for the reduced index set $\mathcal{S} = \mathcal{S}' \setminus \\{i\\}$ is obtained as
>
> $$
> \left({(\boldsymbol{\Sigma_k})}_{\mathcal{S},\mathcal{S}}\right)^{-1}
> = \mathbf{F} - \frac{1}{\omega}\boldsymbol{\gamma}\boldsymbol{\gamma}^{\top}.
> $$
>
> Both updates involve matrix–vector multiplications and outer products, each costing $\mathcal{O}(p_t^2)$. Since $p_t \le p$, the per-update complexity is upper-bounded by $\mathcal{O}(p^2)$. Consequently, the computational cost of the outlier detection step reduces from $\mathcal{O}(KNp^4)$ to $\mathcal{O}(KNp^3)$, and the total complexity of the proposed method becomes $\mathcal{O}(KNp^3 + K^2N)$. We have added this discussion in the revised manuscript.
>
> ---
>
> **References**
>
> [1] J. Sherman and W. J. Morrison, “Adjustment of an inverse matrix corresponding to a change in one element of a given matrix,” *Ann. Math. Stat.*, 1950.
> [2] M. A. Woodbury, “Inverting modified matrices,” *Statistical Research Group, Princeton University*, 1950.
> [3] P. Stoica and R. L. Moses, *Spectral Analysis of Signals*, Prentice-Hall, 2005.
> [4] F. Zhang (Ed.), *The Schur Complement and Its Applications*, Springer, 2005.
> [5] D. A. Harville, *Matrix Algebra from a Statistician’s Perspective*, Springer, 1997.

---

> ### Author Response · Authors · 2025-11-11
> **Response to Reviewer LrAE**
>
> **Comment:**
> *GMMs have been for at least two decades a key milestone and starting point for so many probabilistic modelling problems and challenges. Since the 90s, the EM algorithm stands out as one of the standard solutions, at its easy adaptation to non-Gaussian likelihoods, missing data, and optimization-based learning of parameters have defined a strong SOTA baseline that one should not ignore. In this regard, I see the current proposed work limited due to the lack of comparison and connection with such line of works. The problem introduced is somehow niched to some potential application that is not entirely clear for the reader. Later, one quickly sees the problems that the model could have in three directions: (i) larger dimensionality, (ii) larger rate of outliers, and (iii) consideration of non-Gaussian distributed data.*
>
> **Response (Points (ii) and (iii)):**
> The study focuses on the development of a cell-outlier robust variant of the GMM algorithm by utilizing the FDR-based multiple hypothesis testing procedure for outlier detection; however, the method can be modified or extended to non-Gaussian distributions such as the Student’s $t$ mixture model by maximizing the penalized likelihood defined for the corresponding distribution of the mixture components, and subsequently deriving the test statistics, penalty, surrogate functions, and solutions. We aim to extend the present study in this direction in future work. In the manuscript, we have added text that suggests possible future extensions of the proposed method. We have now extended the results up to $60\\%$ outliers, as presented in the supplementary material. From the figure therein, we observe that the proposed method performs well up to $55\\%$ of cell outliers, after which it breaks down.
>
>
> *(See supplementary figures in [per60ol.pdf](https://drive.google.com/file/d/1iGM4okmXgSl25z4rZN5XKZgsRRGCjy1k/view?usp=sharing))*

---

> ### Author Response · Authors · 2025-11-11
> **Response to Reviewer LrAE**
>
> **Comment:**
> *For my taste as reviewer, the paper is somehow limited on the empirical results, and the exploration of the method performance. The synthetic data presentation is a bit naive with the original parameters and hyperparameters shown in pp. 13 for later discovering them. Whenever the method is relatively simple or easy to compute/analyse, one in the ML/AI community should expect larger experiments and results so the performance is characterised at a max level.*
>
> **Response:**
> We have now performed the experiments with different choices of problem dimension ($p \in \\{5, 10, 25\\}$). We generated the synthetic data for $K \in \\{2,3\\}$ clusters. Each cluster is modeled as a multivariate Gaussian distribution with distinct mean and covariance structures, constructed as follows.
>
> **Cluster Means:**
> For each feature dimension $d \in \\{1, \dots, p\\}$, the mean values across $K$ clusters were assigned as a random permutation of a fixed set of base mean values, ensuring that the cluster centers remain well separated in every dimension:
>
> $$
> \mathbf{M} =
> \begin{bmatrix}
> \boldsymbol{\mu}_1 & \boldsymbol{\mu}_2 & \cdots & \boldsymbol{\mu}_K
> \end{bmatrix}
> \in \mathbb{R}^{p \times K},
> $$
>
> $$
> \mu_{dk} \in
> \begin{cases}
> \mathrm{Perm}\\{5, 25\\}, & K = 2, \\\\
> \mathrm{Perm}\\{5, 20, 35\\}, & K = 3.
> \end{cases}
> $$
>
>
> **Cluster Covariances:**
> For each cluster $k$, a symmetric positive definite covariance matrix $\boldsymbol{\Sigma}_k \in \mathbb{R}^{p \times p}$ was generated as
>
> $$
> \boldsymbol{\Sigma}_k = \mathbf{U}_k \boldsymbol{\Lambda}_k \mathbf{U}_k^{\top},
> $$
>
> where $\mathbf{U}_k$ denotes the eigenvector matrix obtained from the eigendecomposition of a random symmetric matrix
>
> $$
> \mathbf{R}_k = \mathbf{A}_k \mathbf{A}_k^{\top},
> $$
>
> $$
> [\mathbf{A_k}]_{i,j} \sim \mathcal{N}(0,1).
> $$
>
> The diagonal eigenvalue matrix $\boldsymbol{\Lambda_k} = \mathrm{diag}(\lambda_{k1}, \ldots, \lambda_{kp})$ contains independent entries drawn from uniform distributions with cluster-specific spectral ranges:
>
> $$
> \lambda_{ki} \sim
> \begin{cases}
> \mathcal{U}(6, 15), & k = 1, \\\\
> \mathcal{U}(4, 10), & k = 2, \\\\
> \mathcal{U}(3, 8), & k = 3~(\text{if } K=3).
> \end{cases}
> $$
>
> This procedure yields full-rank, well-conditioned covariance matrices with diverse eigenvalue spectra, thereby producing clusters of distinct orientation and spread.
>
> **Sample Generation:**
> For each cluster $k$, data samples were generated as
>
> $$
> \mathbf{X_k} = \boldsymbol{\mu_k} \mathbf{1}_{\pi_k N}^{\top}+ \boldsymbol{\Sigma}_k^{1/2} \mathbf{Z}_k,\qquad
> \mathbf{Z}_k \in \mathbb{R}^{p \times \pi_k N},
> $$
>
> $$
> \mathbf{z}_{k,t} \sim \mathcal{N}(\mathbf{0}, \mathbf{I}_p),
> $$
>
> where each column $\mathbf{z}_{k,t}$ of $\mathbf{Z}_k$ represents an independent $p$-dimensional standard normal vector, $\boldsymbol{\Sigma_k}^{1/2} = \mathbf{U_k} \boldsymbol{\Lambda_k}^{1/2} \mathbf{U_k}^{\top}$,
>
> and $\mathbf{1}_{\pi_k N}$ is a vector of ones of length $\pi_k N$. The total sample count and mixture proportions were set as
>
> $$
> N =
> \begin{cases}
> 400, & K = 2, \\\\
> 600, & K = 3,
> \end{cases}
> \qquad
> \boldsymbol{\pi} =
> \begin{cases}
> [0.5, 0.5], & K = 2, \\\\
> [0.3, 0.33, 0.37], & K = 3.
> \end{cases}
> $$
>
> The results showing the NRMSE for the estimated means and covariances, as well as the KL divergence with respect to the outlier percentage, are presented in the supplementary material, together with tables summarizing the average computation time (in seconds) of each method across different outlier percentages.
>
> *(See supplementary figures and tables in [large_dimension_simulations.pdf](https://drive.google.com/file/d/11YZXiCQ2L18P5jQzQCsanMHjL0UPuPgW/view?usp=sharing))*

---

> ### Author Response · Authors · 2025-11-11
> **Response to Reviewer LrAE**
>
> **Comment:**
> *The experiments and their results shown in Fig. 2 are OK to me, but inevitably remind me to the ones from Figure 1 in Ghahramani & Jordan (1993) [1], where the solution is tested to rates ranging from 1% up to 100%. The way the paper addressed the problem is in many ways similar to such classical method, with EM used to estimate/fill missing data in a robust manner. The lack of discussion and the similarity of the proposed algorithm (taking the definition written in the beginning of section 4.1 pp.5) concern me significantly.*
>
> **Response:**
> We have now extended the experiments to include cases with up to $60\\%$ cell outliers, as detailed in our response to your Comment 1. In addition, we have incorporated a discussion of the work by Ghahramani and Jordan [1] and highlighted the distinctions between their approach and the proposed method in the revised manuscript.
>
> ---
>
> **Reference**
> [1] Z. Ghahramani and M. I. Jordan, “Supervised learning from incomplete data via an EM approach,” *Advances in Neural Information Processing Systems (NeurIPS)*, vol. 6, 1993.

---

> ### Author Response · Authors · 2025-11-11
> **Response to Reviewer LrAE**
>
> **Comment:**
> *The computational cost is correctly characterised, but I am highly concerned about its feasibility in real-world studies or methods with such super-cubical complexity on the vector parameters.*
>
> **Response:**
> As discussed in our response to your Comment 1, the dominant term in the original $\mathcal{O}(KNp^4 + K^2N)$ complexity arises from repeated matrix inversions during outlier detection. By exploiting the fixed covariance structure and applying rank-one update and downdate formulations, the per-iteration cost is reduced by one order to $\mathcal{O}(KNp^3 + K^2N)$. This improvement enhances the scalability of RobGMM for moderately high-dimensional data, making it practical for real-world applications.

---

> ### Author Response · Authors · 2025-11-11
> **Response to Reviewer LrAE**
>
> **Comment:**
> *Last but not least, I miss some references to well-known ML/AI outlier detection methods, and latent variables (that's what it is somehow, but without prior or admitted randomness). The way it is modelled with such masks reminds me of this work from D. Blei's lab some years ago: https://arxiv.org/pdf/1606.03860. I am aware that the scope is quite different, but still useful for the faced problems in the manuscript.*
>
> **Response:**
> We now discuss and cite this reference in the manuscript.

---

> ### Author Response · Authors · 2025-11-11
> **Response to Reviewer LrAE**
>
> **Comment (Requested Changes):**
> *I would like to see additional progress on the main points I raised as weaknesses in my review. From all of theses ones, I would prioritize the EM connection and revision, which is perhaps the usual point-of-reference for readers familiar with GMMs. However, I also want to be honest in the sense that I see the current manuscript significantly under the TMLR threshold, particularly on the interest/fitting side with the potential readers/researchers/practitioners which focus on TMLR published works. In the end, the paper finds a solution, which does not completely scales or adapts to non-Gaussian problems for a GMM problem, well-known in the community and analyzed since the 90s in multiple ways not exactly revised in the paper.*
>
> **Response:**
> In the revised manuscript, we have included a detailed discussion on the connection between the proposed approach and the Expectation–Maximization (EM) framework to better situate the work for the TMLR readership. We note that EM and Majorization–Minimization (MM) yield identical parameter updates $\left(\{\pi_k, \boldsymbol{\mu}_k, \boldsymbol{\Sigma}_k\}\right)$ for the non-outlier case [1]. In the presence of outliers, we prefer MM, as given the latest estimate of outlier positions, EM is typically run until convergence but on the other hand, a single iteration of MM is sufficient to decrease the criterion in (11). Furthermore, we acknowledge the reviewer’s point regarding the extension to non-Gaussian problems and have identified this as a promising future research direction, which is now mentioned explicitly in the revised manuscript.
>
> ---
>
> **Reference**
> [1] N. Sahu and P. Babu, *“New derivation for Gaussian mixture model parameter estimation: MM based approach,”* arXiv preprint arXiv:2001.02923, 2020.

---

> ### Author Response · Authors · 2025-11-11
> **Response to Reviewer LrAE**
>
> **Minor issue (Additional Comments):** *I would've moved the numbers and settings of the numerical studies in pp. 10 and pp. 13 to the appendix, which is the standard practice for venues like TMLR.*
>
> **Response:** Done.

---

### Decision · Action_Editor_mKEX · 2026-01-29

**Recommendation:** Accept as is

**Audience:**

Yes

**Audience Explanation:**

This work proposes a new solution to an problem to which known solutions exist. This will not be a game changer and work is somewhat nice in the context of recent advances in AI, but the methodological contributions are worth sharing with the community.

**Claims And Evidence:**

Yes

**Claims Explanation:**

All reviewers were satisfied with the responses of the authors.This work revisits a long standing problem in statistics/ML, namely robustness against (cell) outliers in the context of GMMs. Several solutions have been proposed in the literature that are able to solve this problem already, but the authors take a different approach. Claims are supported as confirmed by the reviewers, but novelty modest. Following the guidelines of TMLR I recommend acceptance.